# Exploring Feed Digestibility and Broiler Performance in Response to Dietary Supplementation of *Chlorella vulgaris*

**DOI:** 10.3390/ani15010065

**Published:** 2024-12-30

**Authors:** Sofie Van Nerom, Kobe Buyse, Filip Van Immerseel, Johan Robbens, Evelyne Delezie

**Affiliations:** 1Animal Science Unit, Flanders Research Institute for Agriculture, Fisheries and Food (ILVO), 9090 Merelbeke-Melle, Belgium; kobe.buyse@ilvo.vlaanderen.be (K.B.); johan.robbens@ilvo.vlaanderen.be (J.R.); 2Livestock Gut Health Team (LiGHT), Department of Pathobiology, Pharmacology and Zoological Medicine, Faculty of Veterinary Medicine, Ghent University, 9820 Merelbeke-Melle, Belgium; filip.vanimmerseel@ugent.be; 3Department of Veterinary and Biosciences, Faculty of Veterinary Medicine, Ghent University, 9820 Merelbeke-Melle, Belgium

**Keywords:** broiler, *Chlorella vulgaris*, digestibility, intestinal health

## Abstract

This study explored the potential of microalga *Chlorella (C.) vulgaris* as a sustainable protein source in broiler feed. Two trials were conducted to evaluate broilers’ digestion of feeds containing varying levels of *C. vulgaris*. This alga was tested in two forms: unprocessed biomass and biomass processed using a pulsed electric field (PEF), a technique designed to disrupt cell walls. Broiler diets were supplemented with *C. vulgaris* in amounts ranging from 1–20%. The results show decreasing digestibility of protein, fat, and energy as microalgae inclusion levels increased. However, treating *C. vulgaris* with PEF led to improved digestibility of the feed. By exploring alternative protein sources such as microalgae, this study takes a step towards replacement of soybean meal in poultry diets.

## 1. Introduction

The global population is projected to reach 9.7 billion by 2050, driving a significant increase in food and feed demand and the urgent need for alternative protein sources [1]. Currently, soybean meal is a crucial protein source in poultry and livestock diets due to its high quality protein composition. However, sustainability concerns, including deforestation, monocropping, and biodiversity loss, as well as the environmental impacts of conventional crop systems such as greenhouse gas and nitrogen emissions, have raised questions about the long-term viability of soybean meal as a main protein source [2,3]. Possible alternative sources such as peas and other *Leguminosae*, proteins from cereals, insects, and (micro)algae are being explored to address these challenges [4].

Microalgae offer a particularly promising sustainable protein alternative. Cultivated in either bioreactors or raceway ponds, microalgae do not require arable land and exhibit high photosynthetic efficiency, carbon fixation capacity, and rapid growth rates [5]. Protein yields from microalgae production can reach 15 to 30 tons dry matter (DM) ha^−1^ year^−1^ [4].

*Chlorella (C.) vulgaris*, a unicellular, eukaryotic microalga and the cyanobacteria *Arthrospira platensis* are mainly studied for their potential use in feed [6,7,8,9]. *C. vulgaris* is a mixotrophic microalga; it can grow either autotrophically (using light and CO_2_) or heterotrophically (using organic carbon sources) [5]. Both *Chlorella* and *Arthrospira* contain large amounts of high quality proteins (including all essential amino acids), averaging 50 to 60% DM [10]. These microalgae species show potential as a sustainable alternative for soy. Digestibility of feeds including microalgae is a prerequisite for its use as a successful protein alternative in poultry.

Only a few reports of on the effects of microalgae on feed digestibility in animal feed are available [3]. The majority of studies have evaluated the effects of *C. vulgaris* in poultry feed with a focus on its application as a dietary additive due to the presence of bioactive compounds with potential health-promoting effects such as poly-unsaturated fatty acids (PUFAs) and pigments such as β-carotene, lutein, and astaxanthin, which can serve as antioxidants. *C. vulgaris* also contains vitamins (e.g., B1, B2, B3, B5, B6, B7, B9, B12, C, E, and A) and minerals (e.g., Na, K, Ca, Mg, P, Cr, Cu, Zn, Mn, Se, I, and Fe) that have potential health benefits for poultry [5]. The most commonly studied inclusion levels in poultry feed were between 0.01% and 2%, occasionally up to 7.5% [11]. Higher inclusion levels were rarely evaluated [7,12,13]. To the best of our knowledge, no poultry feed digestibility studies with higher inclusion levels of microalgae have been published.

Constraints in digestibility of feeds containing microalgae could be due to a lack of digestibility of the microalgae themselves. Digestibility of *C. vulgaris* may be inhibited due to the specific components of the cell walls: they consist of a three-layer structure formed by compounds such as chitin- or chitosan-like structures, cellulose, hemicellulose, mannan, rhamnose, galactose, uronic acids, glucosamine, and proteins, all of which can make it more difficult to digest because amino acids and other nutrients remain contained within the cells [14,15]. Furthermore, because of the presence of these complex cell wall components, viscosity of the feeds could increase, compromising nutrient uptake [16].

Different techniques are available to break these cell walls, e.g., bead milling (10 kWh kg^−1^) and high-pressure homogenization (0.25 kWh kg^−1^). However, these techniques are energy consuming. Innovative methods such as PEF (0.06 kWh kg^−1^) and ultrasonication (0.07 kWh kg^−1^) are put forward as strategies to reduce energy consumption [3,17]. In previous experiments, a cell disruption efficiency in *C. vulgaris* up to 80% was found using PEF [18]. The main benefit of PEF, compared to the other methods, is that it perforates the cell wall instead of completely breaking it. This may result in less air exposure of nutritional elements such as proteins and lipids, which, in turn, reduces the risk of oxidization.

This research investigated the potential of *C. vulgaris* as a sustainable protein source in broiler feed. It has a high quality protein composition and, thus, represents a promising alternative but digestibility might be compromised due to its rigid cell wall. PEF treatment is investigated for its effect on digestibility of feeds with *C. vulgaris* in broilers. In the current study, digestibility of feeds with autotrophic *C. vulgaris* in broilers was studied at inclusion levels up to 20% (from partially replacing soybean meal to completely replacing soybean meal), both for unprocessed and PEF processed *C. vulgaris*. Linear, quadratic, and broken-line (segmented) regression between microalgae inclusion level and feed digestibility were evaluated to find an optimal inclusion level. Broken-line models are often used to determine maximum safe levels of feed ingredients. Inclusion levels beyond this point can become inefficient or have negative effects on performance [19]. These broken-line models can be used in digestibility trials, to find a maximum point, after which the digestibility decreases at a steeper slope [20,21].

## 2. Materials and Methods

All experiments in this study were performed in compliance with the European guidelines for the care and use of animals in research (Directive 2010/63/EU) and were approved by the Ethics Committee of Flanders Research Institute for Agriculture, Fisheries and Food (ILVO) in Merelbeke-Melle, Belgium under authorization number 2022/446.

### 2.1. Birds and Housing for Two Digestibility Trials (Trial 1 and 2)

A total of 252 Ross 308 one-day-old male broilers were purchased from a commercial hatchery (126 per trial) (Belgabroed, Merksplas, Belgium). The first seventeen days, they were group-housed on a solid floor covered with wood shavings and were fed a basal starter diet (Table 1). A 23 h light/1 h dark light scheme and a room temperature of 32 °C was used during the first week, after which a 18 h light/6 h dark scheme was used for the rest of the rearing period. The temperature of the room was gradually lowered by 4 °C per week until the final temperature of 22 °C in week four. Chickens were vaccinated against Newcastle disease on day fifteen with Nobilis^®^ (Intervet, Boxmeer, Nederland). On day 18, broilers were relocated to digestibility units (L: 0.50 m, W: 0.40 m, H: 0.35 m). One unit with three birds was considered as one replication with a total of six units per treatment. A four-day adaptation period was followed by five consecutive days of balance period (when feed intake (FI) and excreta production are recorded) according to the reference method [22] and as described in [18]. For trial 2, growth and FI was recorded.

### 2.2. Broiler Feed Composition and Nutrient Calculation

Treatments in trial 1 constituted of autotrophic *C. vulgaris* (A) in the following concentrations: 1%, 2%, 5%, 10%, 15%, and 20% (A1, A2, A5, A10, A15, and A20) and a control feed without microalgae (CON).

Treatments in trial 2 constituted of PEF-processed autotrophic *C. vulgaris* (APEF) in the following concentrations: 1%, 2%, 5%, 10%, 15%, and 20% (APEF1, APEF2, APEF5, APEF10, APEF15, and APEF20) and a control feed without microalgae (CON).

To obtain feeds with the correct microalgal inclusion level, four feeds were formulated (F0 with no microalgae, F1 with 1% microalgae, F2 with 2% microalgae, and F20 with 20% microalgae) (Table 1). Feeds F1, F2, and F20 were formulated with either autotrophic algae or PEF-processed autotrophic microalgae. Mixes of these basic feeds were made to obtain feeds with the correct microalgal inclusion level. CON consists of 100% F0; A1 and APEF1 consist of 100% F1; A2 and APEF2 consist of 100% F2; A5 and APEF5 are a mix of 75% F0 and 25% F20; A10 and APEF10 are a mix of 50% F0 and 50% F20; A15 and APEF15 are a mix of 25% F0 and 75% F20; A20 and APEF20 consist exclusively of F20. All feeds were formulated to obtain feeds with equal protein contents.

Broilers were fed a starter diet from day 1 to day 17. From day 18 to day 29, broilers were fed the grower feed including *C. vulgaris*. The autotrophic *C. vulgaris* was purchased from Algademy (Reggio Emilia, Italy). The composition and nutrient analysis of the microalgae are shown in Table 2. Amino acids were analyzed according to [23]. PEF processing of the microalgae was performed according to [18].

### 2.3. Analyzed Feed Composition (Trial 1 and 2)

Feeds were formulated to obtain equal values of protein content, thus crude fat and gross energy decreased as inclusion level of microalgae increased (Table 3). The difference between the 0% and 20% diets is approximately 2.7% in crude fat and 102 kcal in gross energy. Crude protein, crude ash, and crude fiber remained constant for all diets.

### 2.4. Sample Collection, Analysis, and Digestibility Calculations (Trial 1 and 2)

Total FI was determined and all excreta of the five consecutive days of the balance period were collected in collection boxes. The excreta were pooled per unit and weighed. Afterwards, homogenized subsamples from the excreta were freeze-dried, ground, and stored at −20 °C. Excreta were analyzed for gross energy [24], DM (103 °C) [25], crude protein (CP) (N × 6.25) [26], crude fat-B (hydrolysis with HCl followed by extraction with petroleum ether) [27], crude fiber [28], and crude ash [29]. Apparent fecal digestibility coefficients (aFDC) were calculated using the inert marker TiO_2_ (0.4% in the feed) (Equation (1)) for the components crude fiber, crude fat, crude ash, and gross energy. Apparent crude protein digestibility was corrected for the amount of uric acid found in the excreta [30] (Equation (2)). Nitrogen retention was calculated with Equation (3). Metabolizable energy, corrected for nitrogen retention, was calculated with Equation (4) [31].
(1)$$Apparent\;fecal\;digestibility\;coefficient\left(aFDC\right)(\%)=1-\left(\left(\frac{{TiO}_{2\;feed}}{{TiO}_{2\;excreta}}\right)*\left(\frac{{Component}_{excreta}}{{Component}_{feed}}\right)\right)*100\%$$


(2)
$${CP\;digestibility}_{excreta}\;(corrected\;for\;uric\;acid)=\left(\frac{{CP}_{excreta}}{6.25}-\left(\frac{Uric\;acid}{168.1103}*14.0067*4\right)\right)*6.25$$



(3)
$$Nret(J)=\left(\frac{{CP}_{feed}}{6.25}-\left(\frac{{CP}_{excreta}}{6.25}*\frac{{TiO}_{2\;feed}}{{TiO}_{2\;excreta}}\right)\right)*34.36$$



(4)
$$MEn\left(kcal\right)=\frac{\left(\frac{{FDC}_{gross\;energy}\;*\;{Gross\;energy}_{feed}}{100}*4.187\right)-Nret\;(J)}{4.187}$$


### 2.5. Color Measurement of the Breast Filets (Trial 2)

On day 29, the broilers were weighed and euthanized using pentobarbital. Left breast filets of the broilers in trial 2 were weighed and color (Spectrophotometer CM-700d/600d, Konica Minolta, Tokyo, Japan) was measured at three locations on the breast. Color was expressed according to the International Commission on Illumination (CIE) LAB color scale where L* represents lightness on a scale from 0 to 100, a higher number represents a lighter value; a* represents the red–green scale, where a more negative value tends to green and a more positive value tends to red; b* represents the blue–yellow scale, where a more negative value tends to blue, while a more positive value tends to yellow.

### 2.6. Viscosity Determination of Feed and Feces (Trial 1 and 2)

Viscosity of the feed and feces was determined by suspending 2 g of feed in 7 mL distilled water or 1 g of dried feces in 5 mL distilled water. The mixture was shaken for 30 s and subsequently centrifuged for 10 min (feed) or 15 min (feces) at 3000 g and 21 °C. Viscosity of the supernatant was determined with a Viscometer (LVDV2T, Brookfield, Middleboro, MA, USA) with a CP-40 (40) spindle at 40 °C and 0.78 g (feed) and 0.27 g (feces).

### 2.7. Statistical Analysis

Statistical analysis was performed with R version 4.1.2 for Windows [32]. Linear model assumptions (normality and homoscedasticity) were verified by a visual check of the residuals plots. Linear, quadratic, and broken-line models were fitted to the data. Models were considered significant at significance level of α = 0.05 (*p* < 0.05). To evaluate the broken-line models, a Davies test was performed, yielding a *p*-value that indicates whether the model was significantly different from a linear model without a breakpoint [33]. Linear slopes with their confidence intervals (CI) of the models are reported and indicate the change in digestibility with increasing inclusion level. The calculated digestibility coefficients are reported in Table 4 and Table 5. The regression and statistical parameters of the digestibility are shown in Figure 1, Figure 2, Figure 3 and Figure 4.

## 3. Results

### 3.1. Crude Protein Digestibility of A (Trial 1) and APEF (Trial 2)

A decrease in crude protein digestibility was observed in line with increasing microalgae inclusion in both A and APEF feed (Table 4 and Table 5). Both a quadratic and linear relationship were significant (*p* < 0.001) for both A and APEF (Figure 1). The crude protein digestibility of the control feed (0% microalgae inclusion) was 82.04 ± 1.42% in trial 1 and 81.63 ± 1.90% in trial 2. The feeds with 20% microalgae inclusion had a crude protein digestibility of 66.96 ± 1.16% and 72.75 ± 0.34% for A (trial 1) and APEF (trial 2), respectively.

The slope of the linear model for feeds with A (trial 1) was −0.69 [CI: −0.77, −0.61] and the slope of the linear model for feeds with APEF (trial 2) was −0.49 [CI: −0.55, −0.43] (Figure 1). The crude protein digestibility was 78.52 ± 1.18% for the feed with 10% A. The broken-line model shows a breakpoint for the A feed at 10% [CI: 7.25%, 12.75%] inclusion level. The slope of the curve was −0.32 [CI: −0.43, −0.22] and −1.12 [CI: −1.38, −0.87] before and after the breakpoint, respectively (Figure 1).

The Davies test gave a *p*-value of 0.148 for the broken-line model, indicating the change of slopes was not significantly different from the linear slope without a breakpoint. For the feeds with APEF, no broken-line model could be fitted, indicating that there is no inclusion level after which the decrease in digestibility changes.

**Figure 1 animals-15-00065-f001:**
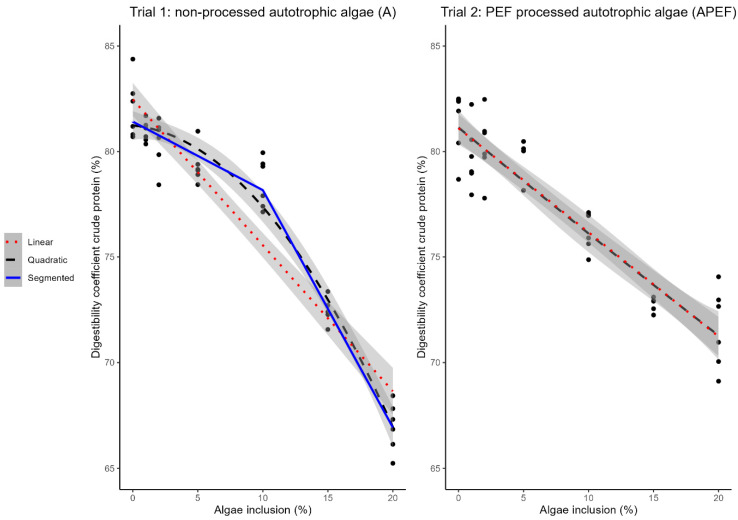
Relation between microalgae inclusion level in the feed and crude protein digestibility. Broken-line model (segmented) (blue, solid line, breakpoint: 10.00% [CI: 7.25%, 12.75%] slope before breakpoint: −0.32 [CI: −0.43, −0.22], slope after breakpoint: −1.12 [CI: −1.38, −0.87], Davies-test: *p*-value: 0.148), linear model (red, dotted line, slope: −0.69 [CI: −0.77, −0.61], *p* < 0.001), and quadratic (black, dashed line, *p* < 0.001) model describing the relationship between crude protein digestibility of the feed (%*w/w* on DM) and inclusion level of unprocessed autotrophic microalgae (A) (left figure, trial 1, n = 6). Linear model (red, dotted line, slope: −0.49 [CI: −0.55, −0.43], *p* < 0.001) and quadratic model (black, dashed line) (*p* < 0.001) describing the relationship between crude protein digestibility of the feed (%*w/w* on DM) and inclusion level of PEF-processed autotrophic microalgae (APEF) (right figure, trial 2, n = 6). Black dots indicate measured data points. PEF: pulsed electric field. CI: confidence interval.

**Table 4 animals-15-00065-t004:** Digestibility parameters and metabolizable energy (mean value ± standard deviation) of the different feed treatments in digestibility trial 1 (unprocessed autotrophic *C. vulgaris*).

Trial 1							
Parameter	CON	A1	A2	A5	A10	A15	A20
Metabolizable energy (n) (kcal/kg)	3059.99 ± 56.62	3073.44 ± 69.55	3112.90 ± 45.24	3050.81 ± 65.44	3048.86 ± 62.91	2733.63 ± 183.17	2629.42 ± 57.64
Gross energy (%)	75.56 ± 1.40	75.66 ± 1.68	76.63 ± 1.14	75.10 ± 1.59	74.99 ± 1.51	67.14 ± 4.40	64.50 ± 1.36
Crude fat (%)	88.01 ± 2.39	88.14 ± 1.36	87.07 ± 1.96	84.75 ± 3.06	82.84 ± 2.63	73.44 ± 3.82	57.80 ± 5.99
Crude protein (%)	82.04 ± 1.42	80.95 ± 0.50	80.45 ± 1.15	79.33 ± 0.86	78.52 ± 1.18	72.46 ± 0.65	66.96 ± 1.16
Crude ash (%)	34.43 ± 1.53	34.77 ± 1.31	33.82 ± 1.40	31.95 ± 2.23	28.87 ± 1.48	25.66 ± 1.49	25.06 ± 2.61
Crude fiber (%)	0.00	0.00	0.00	0.00	0.00	0.00	0.00

A1–20: autotrophic *C. vulgaris* from 1 to 20% inclusion level, CON: control. n: apparent metabolizable energy (nitrogen corrected) of the feed.

**Table 5 animals-15-00065-t005:** Digestibility parameters and metabolizable energy (mean value ± standard deviation) of the different feed treatments in digestibility trial 2 (PEF processed autotrophic *C. vulgaris*).

Trial 2							
Parameter	CON	APEF1	APEF2	APEF5	APEF10	APEF15	APEF20
Metabolizable energy (n) (kcal/kg)	3123. 64 ± 28.12	3081.31 ± 44.76	3112.21 ± 26.66	3100.07 ± 48.63	3022.63 ± 25.60	2957.96 ± 42.63	2905.81 ± 63.34
Gross energy (%)	77.50 ± 0.73	76.39 ± 1.13	77.21 ± 0.66	76.91 ± 1.29	74.76 ± 0.63	73.19 ± 1.07	71.80 ± 1.49
Crude fat (%)	89.71 ± 1.39	87.80 ± 2.34	87.84 ± 2.75	88.06 ± 1.76	82.92 ± 1.65	81.20 ± 1.64	73.46 ± 1.30
Crude protein (%)	81.63 ± 1.90	81.39 ± 1.54	79.75 ± 1.50	80.29 ± 1.57	79.70 ± 2.98	76.25 ± 0.92	72.75 ± 0.34
Crude ash (%)	33.16 ± 2.12	32.55 ± 1.79	33.91 ± 1.83	33.09 ± 2.80	28.33 ± 2.16	25.51 ± 2.68	26.06 ± 2.54
Crude fiber (%)	0.00	0.00	0.00	0.00	0.00	0.00	0.00

APEF1–20: autotrophic *C. vulgaris* from 1 to 20% inclusion level, CON: control, PEF: pulsed electric field. n: apparent metabolizable energy (nitrogen corrected) of the feed.

### 3.2. Crude Fat Digestibility of A (Trial 1) and APEF (Trial 2)

Both a quadratic and linear relationship were highly significant (*p* < 0.001) for both A and APEF (Figure 2). Increasing microalgae inclusion level in the feeds led to a decrease in crude fat digestibility, both for A and APEF (Table 4 and Table 5). The crude fat digestibility of the control feed (0% microalgae inclusion) was 88.01 ± 2.39% in trial 1 and 89.71 ± 1.39% in trials 1 and 2, respectively. The 20% microalgae feed had a crude fat digestibility of 57.80 ± 5.99% and 73.46 ± 1.30% for feeds including A and APEF, respectively.

The slope of the linear model for feeds with A and APEF are −1.39 [CI: −1.61, −1.17%], and −0.65 [CI: −0.77, −0.53], respectively (Figure 2). The broken-line model shows a breakpoint for the A feed at 12.53% [CI: 10.41%, 14.65%] inclusion level. The slope of the curve before and after this breakpoint was −0.56 [CI: −0.88, −0.23] and −3.13 [CI: −3.91, −2.35], respectively (Figure 2).

The Davies test gave a *p*-value of 0.930 for the broken-line model, indicating the change in slopes was not significantly different from the linear slope without breakpoint. For the feeds including APEF, no broken-line model could be fitted.

**Figure 2 animals-15-00065-f002:**
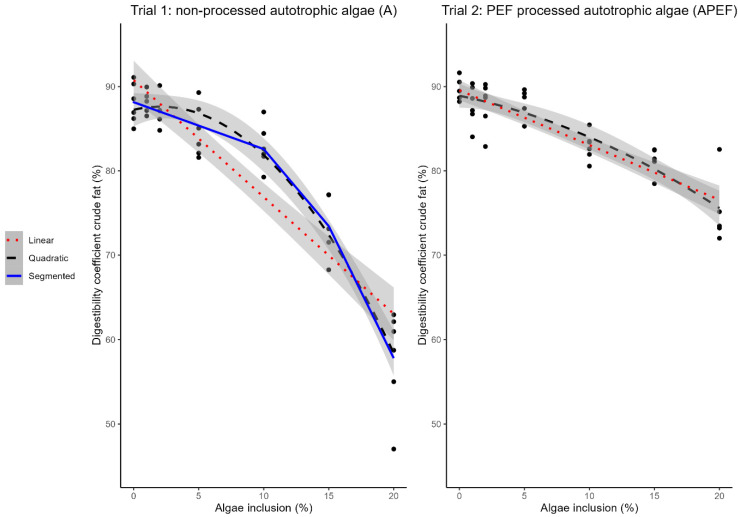
Relation between microalgae inclusion level in the feed and crude fat digestibility. Broken-line model (segmented) (blue, solid line, breakpoint: 12.53% [CI: 10.41%; 14.65%], slope before breakpoint: −0.56 [CI: −0.88, −0.23], slope after breakpoint: −3.13 [CI: −3.91, −2.35], Davies-test: *p*-value: 0.930), linear model (red, dotted line, slope: −1.39 [CI: −1.61, −1.17], *p* < 0.001), and quadratic model (black, dashed line) (*p* < 0.001) describing the relationship between crude fat digestibility of the feed (%*w*/*w* on DM) and inclusion level of unprocessed autotrophic microalgae (A) (left figure, trial 1, n = 6). Linear model (red, dotted line, slope: −0.65 [CI: −0.77, −0.53], *p* < 0.001) and quadratic model (black, dashed line, *p* < 0.001) describing the relationship between crude fat digestibility of the feed (%*w*/*w* on DM) and inclusion level of PEF-processed autotrophic microalgae (APEF) (right figure, trial 2, n = 6). Black dots indicate measured data points. PEF: pulsed electric field. CI: confidence interval.

### 3.3. Gross Energy Digestibility of A (Trial 1) and APEF (Trial 2)

A decrease in gross energy digestibility was observed with increasing microalgae inclusion level in both A and APEF (Table 4 and Table 5). Both the linear and quadratic relationship were highly significant (*p* < 0.001) for both A and APEF (Figure 3). The gross energy digestibility of the control feed (0% microalgae inclusion) was 75.56 ± 1.40% in trial 1 and 77.50 ± 0.73% in trials 1 and 2, respectively. The 20% microalgae feed had a gross energy digestibility of 64.50 ± 1.36% and 71.80 ± 1.49% for feeds with A and APEF, respectively.

The slope of the linear model was −0.58 [CI: −0.70, −0.45] and −0.28 [CI: −0.33, −0.28] for feeds including A and APEF, respectively (Figure 3). A broken-line model gave a breakpoint for the A feed at 9.26% [CI: 4.98%, 13.54%] inclusion level. The slope of the curve before this breakpoint was −0.10 [CI: −0.59, 0.39] and −1.07 [CI: −1.34, −0.79] before and after the breakpoint, respectively (Figure 3). The gross energy digestibility was 74.99 ± 1.51% for the feed with 10% A.

**Figure 3 animals-15-00065-f003:**
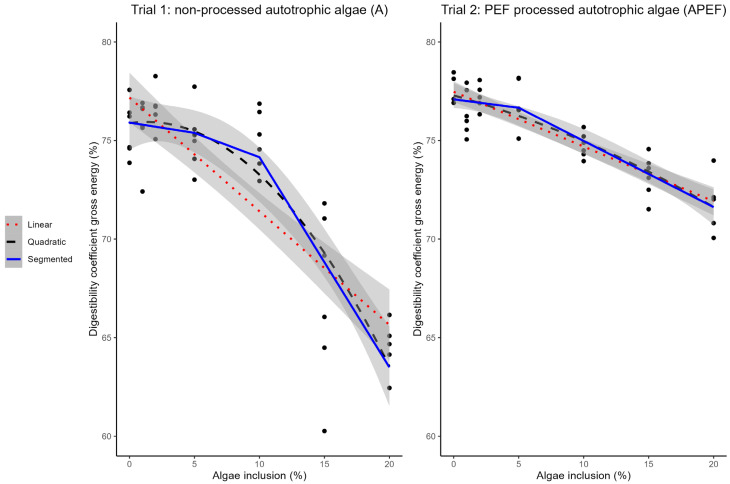
Relation between microalgae inclusion level in the feed and gross energy digestibility. Broken-line model (segmented) (blue, solid line, breakpoint: 9.26% [CI: 4.98%, 13.54%], slope before breakpoint: −0.1 [CI: −0.59, 0.39], slope after breakpoint: −1.07 [CI: −1.34, −0.79], Davies-test: *p*-value: 0.125), linear model (red, dotted line, slope: −0.58 [CI: −0.70, −0.45], *p* < 0.001), and quadratic model (black, dashed line, *p* < 0.001) describing the relationship between gross energy digestibility of the feed (%w/w on DM) and inclusion level of unprocessed autotrophic microalgae (A) (left figure, trial 1, n = 6). Broken-line model (segmented) (blue, solid line, breakpoint: 5 [CI: −6.11, 16.11], slope before breakpoint: −0.09 [CI: −0.74, 0.57], slope after breakpoint: −0.34 [CI: −0.42, −0.25], Davies-test: *p*-value: 0.641), linear model (red, dotted line, slope: −0.28 [CI: −0.33, −0.28], *p* < 0.001), and quadratic model (black, dashed line, *p* < 0.001) describing the relationship between gross energy digestibility of the feed (%w/w on DM) and inclusion level of PEF-processed autotrophic microalgae (APEF) in the feed (%) (right figure, trial 2, n = 6). Black dots indicate measured data points. PEF: pulsed electric field. CI: confidence interval.

The Davies test gave a *p*-value of 0.125 for the broken-line model, indicating the change in slopes was not significantly different from the linear slope without breakpoint. A broken-line model gave a breakpoint for the APEF feed at 5% [CI: −6.11%, 16.11%] inclusion level. The slope of the curve before this breakpoint was −0.09 [CI: −0.74, 0.57] and −0.34 [CI: −0.42, −0.25] before and after the breakpoint, respectively (Figure 3). The gross energy digestibility was 76.91 ± 1.29% for the feed with 10% APEF. The Davies test gave a *p*-value of 0.641 for the broken-line model.

### 3.4. Crude Ash Digestibility of A (Trial 1) and APEF (Trial 2)

A decrease in crude ash digestibility was observed with increasing levels of microalgae inclusion for both A and APEF (Table 4 and Table 5). Both a quadratic and linear relationship showed highly significant (*p* < 0.001) differences for both A and APEF (Figure 4). The crude ash digestibility of the control feed (0% microalgae inclusion) was 34.43 ± 1.53% in trial 1 and 33.16 ± 2.12% in trials 1 and 2, respectively. The 20% microalgae feed had a crude ash digestibility of 25.06 ± 2.61% and 26.06 ± 2.54% for feeds with A and APEF, respectively.

The slope of the linear model was −0.53 [CI: −0.61, −0.45] and −0.44 [CI: −0.55, −0.34] for feeds with A and APEF, respectively. (Figure 4). No broken-line models could be fitted for either trial.

**Figure 4 animals-15-00065-f004:**
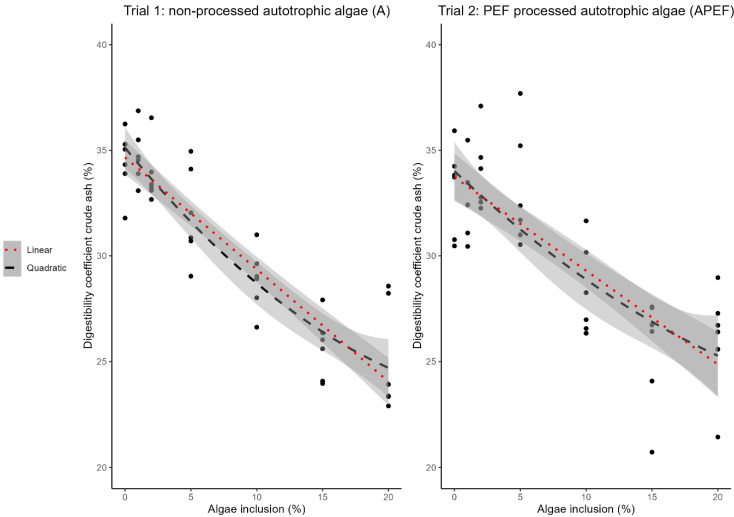
Relation between microalgae inclusion level in the feed and crude ash digestibility. Linear model (red, dotted line, slope: −0.53 [CI: −0.61, −0.45], *p* < 0.001) and quadratic model (black, dashed line, *p* < 0.001) describing the relationship between crude ash digestibility of the feed (%w/w on DM) and inclusion level of autotrophic microalgae (A) (left figure, trial 1, n = 6). Linear model (red, dotted line, slope: −0.44 [CI: −0.55, −0.34], *p* < 0.001) and quadratic model (black, dashed line) (*p* < 0.001) describing the relationship between crude ash digestibility of the feed (%w/w on DM) and inclusion level of PEF-processed autotrophic microalgae (APEF) (right figure, trial 2, n = 6). Black dots indicate measured data points. PEF: pulsed electric field. CI: confidence interval.

### 3.5. Growth, Feed Intake, Breast Weight, and Meat Color in Trial 2

Table 6 shows the final body weight (BW), average daily gain (ADG), and average daily FI (ADFI) of the broilers fed with PEF-processed autotrophic microalgae during the observation period. A significant linear decrease in BW (*p* < 0.001), ADG (*p* < 0.001) and ADFI (*p* = 0.006) was observed as microalgae inclusion level increased.

The left breast absolute and relative weight at day 29 is also shown in Table 6. Again, a significant linear decreasing relationship was found (*p* < 0.001) between absolute and relative weight of the breast filets and increasing microalgae inclusion level. The color parameters also changed significantly with increasing microalgae inclusion level. L* shows a significant linear decrease (*p* = 0.029), while b* and a* show a significant linear increase (*p* < 0.001) with increasing microalgae inclusion level (Table 6).

**Table 6 animals-15-00065-t006:** Average daily gain (ADG) in gram per day per animal (g/d/a) (n = 6), average daily feed intake (ADFI) in gram per day per animal (g/d/a), left breast weight (g), and color parameters (L* a* b*) of the breast on day 29 (n = 18) and of broilers fed a diet with a specified inclusion level of PEF-processed autotrophic *C. vulgaris* during five days (balance period d22–d26) and final body weight (BW) on day 26 (trial 2).

Trial 2										
Parameter	CON	APEF1	APEF2	APEF5	APEF10	APEF15	APEF20	Slope (Linear)	CI	*p*-Value
BW (g)	969.8 ± 47.1	918.6 ± 107.9	895.5 ± 41.3	911.2 ± 63.3	833.5 ± 30.5	830.6 ± 54.3	779.6 ± 65.5	−8.00	−10.73, −5.26	<0.001
ADG (g/d/a)	66.5 ± 14.1	56.0 ± 23.1	63.5 ± 11.9	57.4 ± 15.2	41.1 ± 7.67	46.2 ± 10.6	34.1 ± 24.8	−1.43	−2.13, −0.73	<0.001
ADFI (g/d/a)	410.8 ± 29.3	423.5 ± 50.9	400.5 ± 28.1	403.0 ± 39.0	373.9 ± 25.9	386.1 ± 17.2	375.7 ± 30.5	−2.02	−3.44, −0.61	0.006
Breast weight (g)	107.3 ± 17.2	103.0 ± 18.7	95.8 ± 19.1	98.5 ± 18.5	91.8 ± 13.3	90.0 ± 14.3	78.7 ± 12.9	−1.15	−1.55, −0.74	<0.001
Relative breast weight (%)	8.1 ± 0.8	8.0 ± 0.4	7.5 ± 0.9	7.7 ± 0.8	7.7 ± 0.7	7.4 ± 0.7	7.0 ± 0.8	−0.04	−0.06, −0.02	<0.001
L*	56.0 ± 1.5	56.3 ± 1.8	55.9 ± 0.9	55.3 ± 1.8	54.9 ± 1.1	54.2 ± 1.4	55.2 ± 1.8	−0.07	−0.14, −0.01	0.029
a*	0.1 ± 1.0	0.0 ± 0.6	1.0 ± 0.6	0.9 ± 1.3	2.6 ± 0.8	2.1 ± 1.1	2.3 ± 1.2	0.11	0.06, 0.16	<0.001
b*	12.1 ± 1.0	13.6 ± 0.7	15.8 ± 0.9	18.3 ± 0.7	19.9 ± 1.1	20.3 ± 1.5	21.2 ± 1.7	0.41	0.33, 0.49	<0.001

Mean values ± standard deviation and slopes of the linear correlation with their *p*-values. APEF1–20: autotrophic *C. vulgaris* from 1 to 20% inclusion level, PEF: pulsed electric field, CI: confidence interval of the slope.

### 3.6. Viscosity and Water Content of the Feces in Trials 1 and 2

A linear increase in viscosity (*p* < 0.001) and a linear decrease in water content (*p* < 0.001) of the feces was observed in both trials with increasing inclusion level of autotrophic *C. vulgaris* (Table 7). A linear increase in viscosity (*p* < 0.001) of the feed with increasing inclusion level of microalgae was also found.

## 4. Discussion

Microalgae are primarily studied for their potential health-promoting effects at low dietary inclusion levels (typically ranging from 0.01% to 5%). Most studies focus on the impact of microalgae inclusion on performance and health, rather than on the digestibility of the microalgae itself [12,13,34]. Only a few studies on the digestibility of *C. vulgaris* have been published, all of which evaluate low inclusion levels. Panaite et al. (2023) reported a significant decrease in crude fat digestibility at 2% *C. vulgaris* inclusion level in a laying hen diet, whereas crude protein and organic matter digestibility were not affected. Inclusion of 2% *A. platensis* did not show a significant difference in crude fat digestibility [35]. The difference between the effects of the different microalgae might be explained by the rigid cell wall of *C. vulgaris* [15].

Only few digestibility studies on other microalgae (e.g., *Nannochloropsis* sp., *Dunaliella* sp. and *Chloromonas* sp.) have been performed. Tavernari et al. (2018) studied the effects of 20% *A. platensis* inclusion in broiler feed on apparent nitrogen-corrected metabolizable energy and found no significant difference [36]. However, since *A. platensis* lacks the rigid cellulose cell wall found in *C. vulgaris*, digestibility is less compromised for this species of microalgae [37]. The cell wall of *A. platensis* consists mainly of peptidoglycan and lipopolysaccharides [38]. Pestana et al. (2020) studied the potential of adding 0.01% lysozyme to the feed to improve the effects of *A. platensis* [39]. They found that overall broiler performance was reduced, possibly due to the increased viscosity of the digesta, which might be caused by indigestible proteins and polysaccharides. The addition of enzymes that can break down the rigid cell wall of *C. vulgaris* may offer a solution to improve the availability and digestibility of nutrients within the cells. In the current study, non- starch polysaccharides (NSP) enzymes are included, as is common in standard chicken diets. This could also impact digestibility, as these enzymes may be able to break down the polysaccharide cell wall of *C. vulgaris*.

Nevertheless, given the high protein content of microalgae, especially *C. vulgaris*, application of higher dosages may be interesting as part of the ongoing search for alternative protein sources.

In the present study, the effects of microalgae inclusion on feed digestibility were examined. Feeds including unprocessed autotrophic microalgae indicated a steeper decrease in digestibilities of crude protein, crude fat, and gross energy at increasing microalgae inclusion level than feeds with PEF-processed autotrophic microalgae. A more pronounced decrease in crude ash digestibility of feeds with autotrophic microalgae was found in comparison to feeds with PEF-processed autotrophic microalgae, although the observed overlap in the 95% confidence intervals of the linear slopes indicates uncertainty. For instance, when analyzing crude protein, linear slopes of −0.69 and −0.49 were found for unprocessed and PEF-processed autotrophic microalgae, respectively. This indicates that at 10% inclusion level, the digestibility coefficient has already decreased by 6.9% and 4.9% for A and APEF, respectively, showing that the digestibility of unprocessed microalgae decreases faster than that of PEF-processed microalgae. In general, for the unprocessed autotrophic microalgae, a steeper decrease in nutrient digestibility was observed after a breakpoint of 10% compared to the decrease before this breakpoint. For crude protein digestibility, the breakpoint occurred at 10%. For crude fat digestibility, it occurred at 12.5% and for gross energy digestibility it occurred at 9.26%. Furthermore, a breakpoint was also found for gross energy digestibility in the diet including 5% PEF-processed autotrophic microalgae. However, as the feeds were formulated based on equal crude protein content, higher inclusion levels also led to lower amounts of crude fat and energy in the feeds. This might also partly explain the lower digestibility of crude fat and gross energy. On the other hand, a lower amount of fat and energy in the feed could also lead to an increased digestibility, since the nutrients are more scarce and the birds would need to use all available nutrients. Nevertheless, the decreasing digestibility correlation was observed for crude protein and crude ash, while all feeds contained similar amounts of these nutrients. It was expected that replacement of soybean meal, with its optimal protein composition, with microalgae would lead to a decrease in digestibility at higher microalgae inclusion levels.

Conventional broiler feeds contain approximately 20% soybean meal with a crude protein content of 48%. It would require a 20% inclusion level of *C. vulgaris* to meet the crude protein requirements in feed, since *C. vulgaris* as used in this study contained approximately 50% crude protein. The results of this study indicate that inclusion at 20% of *C. vulgaris* in the feed are contraindicated, thus only a partial replacement of the conventional soybean meal would be possible under these circumstances. The decreasing linear correlations and breakpoints found in this study show that inclusion levels over 10% are not recommended. However, even at 10% inclusion, digestibility of crude ash already dropped by approximately 5%. For crude protein in the PEF-processed autotrophic microalgae feeds, the digestibility at 10% is also already 5% lower in comparison to the control group. In terms of gross energy digestibility, inclusion in excess of 5% may be problematic.

Impaired nutrient digestibility at higher doses could be attributed to the complex and rigid cell wall of *C. vulgaris*. The cell wall contains cross-linked insoluble carbohydrates such as chitin- and chitosan-like polymers and cellulose [15]. This might compromise the digestibility in monogastric animals [40]. Different strains of *C. vulgaris* can differ in their cell wall composition. Other (monosaccharide) components reported in *C. vulgaris*’ cell walls are rhamnose, galactose, glucosamine, arabinose, fucose, glucose, uronic acids, and 6–10% proteins [15]. Several techniques have been reported that could improve the digestibility of *C. vulgaris* by disrupting or breaking cell walls. Enzymatic disruption can be accomplished for example using lysozymes [41]. Canelli et al. (2021) found an increase in protein bio-accessibility after enzyme (chitinase, rhamnohydrolase, and galactanase) treatment [42]. After this treatment, the oxidative stability was maintained, compared to the treatment after high-pressure homogenization that led to the formation of off-flavors. Kose et al. (2017) found an increase in *C. vulgaris* in vitro digestibility from 35% to 70% after a pancreatin hydrolysis [43]. Supplementing enzymes to the feed might enhance *C. vulgaris* digestibility. Physical techniques, such as sonication, high-pressure homogenization, and bead milling, are available and may increase digestibility. In the present trial, PEF processing was used to increase the bio-accessibility of nutrients. In a previous study, a disruption efficiency up to 80% of *C. vulgaris* cells using PEF was found [18]. PEF only disrupts the cell wall by perforation, maintaining the cells’ circular shape. PEF treatment might be insufficient under higher inclusion levels of *C. vulgaris* in feeds. However, PEF was chosen in this study as perforation of the cells is preferred over cell breakage because perforation helps to prevent oxidation of the nutrients. More severe techniques, such as high-pressure homogenization or bead milling, could break the cells in smaller fragments to make the nutrients even more available to intestinal enzymes. In the current trial, an increase in viscosity and decrease in water content was found along with increasing microalgal inclusion level. High viscosities can indeed lead to negative effects on nutrient digestibility [44]. When higher inclusion levels of *C. vulgaris* are added in poultry feed, higher dosages of NSP enzymes might be required to be able to digest all the *C. vulgaris* cell walls, thus maximizing nutrient availability. The composition of the NSP enzymes used is also important. NSP enzymes can include xylanase, which will have an effect on the *C. vulgaris* cell wall. Furthermore, Bleakley and Hayes (2017) suggested that the high amounts of polysaccharides might be the main reason for compromised protein digestibility [45]. Another reason might be the presence of phenolic compounds that can bind with amino acids to form insoluble structures.

Complete digestion of *C. vulgaris* cells is important, as full availability of digestible amino acids (AA) is crucial in broiler diets. Feed formulations have specific requirements for different amino acids depending on the broiler’s growth phase. Methionine, lysine, and threonine are often limited in feed and are often supplemented synthetically [46]. *C. vulgaris* contains all essential AA with a notably high lysine content. The amino acid composition can vary, with lysine, arginine, threonine, leucine, and methionine present in notable amounts. However, several studies report significant variance in AA composition of *C. vulgaris* due to differences in strain and growth conditions (e.g., nutrients in growth medium and light) [47,48]. In soy, the amino acid composition remains stable among varieties, which makes it reliable for the use in feed [49]. However, by keeping the growth conditions of a specific type of *C. vulgaris* constant, such reliability is also within reach for microalgae biomass. Figure 5 shows the amino acid composition of the *C. vulgaris* used in the current study and soybean meal 48 (as reported by CVB (2018) [50]). This comparison clearly shows that for most of the amino acids except alanine and glycine, the percentage in soybean meal is higher than in *C. vulgaris*. However, inclusion of *C. vulgaris* as a protein source might have other beneficial effects for broilers due to the antioxidative and immunostimulant capacity of the bio-active compounds present in *C. vulgaris* [51].

In the present study, reduced digestibility of feeds with increasing amounts of *C. vulgaris* also led to decreased FI, which appeared to be related to a decrease in ADG. Several articles suggested reduced palatability of feeds containing microalgae [52,53]. Furthermore, high amounts of polysaccharides in the microalgae feeds can cause faster satiety, thus leading to decreased FI and growth. In addition to the effects on performance, meat quality was also affected by the increasing amount of *C. vulgaris* in the feeds. The CIELAB values in the present study indicated that the breast filets became more dark, yellow and red. Similarly, Altmann et al. (2018) found that inclusion of 10% *A. platensis* led to darker, more red and yellow breasts, but increased meat quality, as the pH was higher, which, in turn, led to less cooking loss [8]. Fillets were more tender and soft and had less metallic off-taste. Furthermore, microalgae can enrich meat with omega n-3 fatty acids, since they are high in PUFAs [53].

Inclusion of *C. vulgaris* in broiler feeds will require techniques that are effective in breaking cells and fragmentation of cell wall components to reduce viscosity of intestinal content. Although such treatment may result in decreased oxidative stability of the nutrients. This study showed reduced digestibility with increasing inclusion levels, leading to reduced performance. Therefore, focusing on the application of *C. vulgaris* as feed additive for its potential antioxidant and prebiotic effects may be an interesting path for future research.

## 5. Conclusions

This study examined the digestibility of broiler feeds including autotrophic *C. vulgaris* by adding unprocessed and PEF-processed microalgae in the feed at inclusion levels up to 20%. Digestibility of the feeds was reduced at increasing inclusion levels of *C. vulgaris*. PEF processing of autotrophic *C. vulgaris* mitigated these effects on digestibility.

Broken-line models identified critical thresholds for unprocessed *C. vulgaris*: 10% for crude protein, 12.53% for crude fat, and 9.26% for gross energy. No significant breakpoints were observed for PEF-processed microalgae except for gross energy, indicating a more gradual decline in digestibility.

Limiting *C. vulgaris* inclusion to no more than 10% of the diet is advisable, although digestibility still falls short compared to soybean meal, particularly for crude protein and ash. While *C. vulgaris* is a promising sustainable protein source, further research into processing methods and feed formulations (e.g., adding enzymes) is needed to enhance its suitability for broiler diets.

## Figures and Tables

**Figure 5 animals-15-00065-f005:**
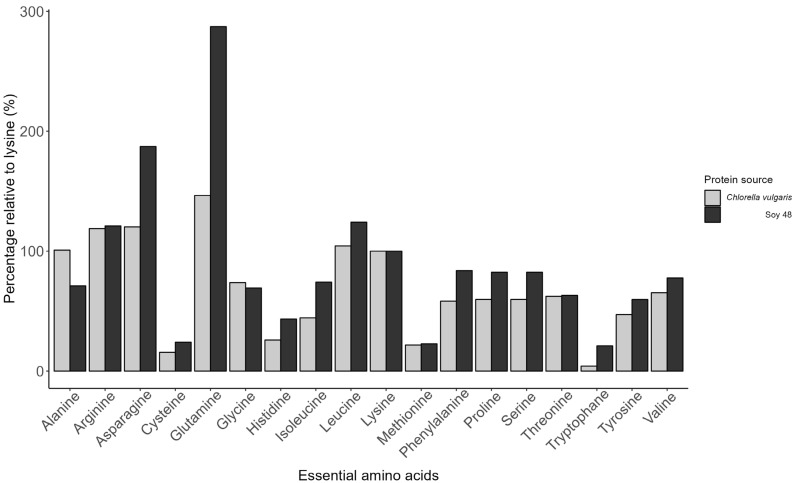
Essential amino acids levels relative to lysine (%) of *C. vulgaris* used in the current study and of soybean meal 48 reported by CVB, (2018).

**Table 1 animals-15-00065-t001:** Feed composition (%) and calculated nutrient composition (%) for the starter feed and basic feeds (F) with 0, 1, 2, and 20% microalgae included. The control feed (CON) is F0, A1 and APEF1 are F1, A2 and APEF2 are F2, A5 and APEF5 are a mix of 75% F0 and 25% F20, A10 and APEF10 are a mix of 50% F0 and 50% F20, A15 and APEF15 are a mix of 25% F0 and 75% F20, A20 and APEF20 are F20.

	Starter	F0	F1	F2	F20
**Ingredient (%)**					
Wheat	56.17	51.67	52.45	53.23	62.37
Wheat bran	0	0	0	0	10.00
Maize	10.00	10.00	10.00	10.00	1.32
Soybean	5.00	7.26	7.00	7.00	0
Soybean meal (48% CP)	22.99	22.48	21.32	19.94	0
Soy oil	0.89	1.00	1.00	1.00	0
Animal fat	1.00	4.00	3.64	3.22	2.50
Mineral and vitamin premix ^1^	1.00	1.00	1.00	1.00	1.00
Feed chalk	0.76	0.68	0.67	0.67	0.50
Di-calcium phosphate	0.74	0.67	0.69	0.72	1.23
NaCl	0.13	0.16	0.16	0.16	0.13
Na-bicarbonate	0.29	0.24	0.25	0.25	0.30
L-lysine HCl	0.38	0.28	0.29	0.29	0.31
DL-methionine	0.31	0.27	0.26	0.26	0.09
L-threonine	0.16	0.12	0.12	0.12	0.09
Coccidiostat	0.05	0.05	0.05	0.05	0.05
NSP enzyme	0.01	0.01	0.01	0.01	0.01
Phytase	0.10	0.10	0.10	0.10	0.10
Autotrophic (PEF) *C. vulgaris*	0	0	1.00	2.00	20.00
**Calculated nutrient composition**					
Crude protein (%)	20.50	20.40	20.40	20.40	20.40
Crude fat (%)	5.87	9.35	9.01	8.67	6.59
Crude ash (%)	4.59	4.46	4.50	4.54	5.38
Crude fiber (%)	3.36	3.21	3.20	3.20	3.67
Metabolizable energy (kcal/kg)	2770	2928	2928	2928	2943
Dig. lysine (%)	1.14	1.08	1.08	1.08	1.08
Dig. methionine + cysteine (%)	0.86	0.81	0.81	0.81	0.81
Ca (%)	0.85	0.80	0.80	0.80	0.80
Available P (%)	0.47	0.46	0.46	0.45	0.44
NaCl + KCl (mEq/kg)	232.13	234.32	227.40	220.47	107.52

^1^ Mineral and vitamin premix composed of vitamin A/retinyl acetate 3a672a (1,000,000 IU kg^−1^); vitamin D3 E671 (299,999.4 IU kg^−1^); vitamin E 3a700 (all-rac-alpha-tocopheryl acetate) (5000 IU kg^−1^); vitamin K3 3a710 (250 mg kg^−1^); vitamin B1/thiamine mononitrate 3a821 (200 mg kg^−1^); vitamin B2/riboflavin (500 mg kg^−1^); calcium D-pantothenate 3a841 (1500 mg kg^−1^); vitamin B6/pyridoxine hydrochloride 3a831 (400 mg kg^−1^); vitamin B12/cyanocobalamine (2.5 mg kg^−1^); niacinamide 3a315 (3000 mg kg^−1^); folic acid 3a316 (100 mg kg^−1^); biotin/D-(+)-biotin 3a880 (15 mg kg^−1^); choline chloride 3a890 (68,965.5 mg kg^−1^); iron (II) sulphate (monohydrate)—iron E1 (4920 mg kg^−1^); copper (II) sulphate (pentahydrate)—copper E4 (2000 mg kg^−1^); zinc oxide 3b603 (6000 mg kg^−1^); manganese (II) oxide—manganese E5 (9590.2 mg kg^−1^); calcium iodate (anhydrous)—iodine 3b202 (120 mg kg^−1^); sodium selenite—selenium E8 (36 mg kg^−1^); sepiolite E562 (700 mg kg^−1^); propyl gallate E310 (200 mg kg^−1^); BHT E321 (300 mg kg^−1^); citric acid E330. PEF: pulsed electric field, A: autotropic *C. vulgaris*, APEF: PEF-processed autotrophic *C. vulgaris*, CP: crude protein, NSP: non-starch polysaccharides.

**Table 2 animals-15-00065-t002:** Composition of the autotrophic *C. vulgaris* on fresh weight.

Parameter	Autotrophic *C. vulgaris*
Gross energy (kcal/kg)	4838
Rest fraction water (%)	5.57
Crude protein (%)	53.75
Crude fat (%)	8.30
Crude ash (%)	9.57
Crude fiber (%)	2.17
**Amino acids (%)**	
Alanine	3.43
Arginine	4.04
Sum of asparagine and aspartic acid	4.09
Cysteine	0.53
Sum of glutamine and glutamic acid	4.98
Glycine	2.51
Histidine	0.88
Isoleucine	1.51
Leucine	3.55
Lysine	3.40
Methionine	0.74
Phenylalanine	1.98
Proline	2.03
Serine	2.03
Threonine	2.12
Tyrosine	1.60
Tryptophan	0.14
Valine	2.22

**Table 3 animals-15-00065-t003:** Analyzed nutrient composition on fresh weight basis of the feeds in trial 1 (i.e., inclusion of unprocessed autotrophic *C. vulgaris)* and trial 2 (i.e., inclusion of PEF-processed autotrophic *C. vulgaris*).

Trial 1							
Analyzed Nutrient Composition	CON	A1	A2	A5	A10	A15	A20
Gross energy (kcal/kg)	4270	4243	4223	4237	4184	4174	4169
Rest fraction water (%)	9.54	9.64	9.56	9.26	9.27	9.05	8.67
Crude protein (%)	20.53	19.96	20.05	19.92	19.48	19.68	19.99
Crude fat (%)	8.61	8.22	7.88	7.74	7.15	6.53	5.89
Crude ash (%)	5.16	5.28	5.25	5.56	5.73	5.86	5.97
Crude fiber (%)	2.98	2.79	2.70	2.90	3.00	2.94	3.08
**Trial 2**							
**Analyzed Nutrient Composition**	**CON**	**APEF1**	**APEF2**	**APEF5**	**APEF10**	**APEF15**	**APEF20**
Gross energy (kcal/kg)	4241	4229	4216	4214	4177	4137	4137
Rest fraction water (%)	9.70	9.65	9.71	9.61	9.33	8.91	8.67
Crude protein (%)	19.90	19.94	20.07	19.98	19.43	19.55	19.41
Crude fat (%)	8.28	8.08	7.35	7.65	6.85	6.29	5.59
Crude ash (%)	5.23	5.29	5.36	5.41	5.36	6.05	5.87
Crude fiber (%)	2.96	2.77	2.80	3.10	2.83	2.74	2.92

PEF: pulsed electric field, CON: control, A1–20: feed composition including 1–20% of unprocessed autotrophic *C. vulgaris*. APEF1–20: feed composition including 1–20% of PEF-processed autotrophic *C. vulgaris*.

**Table 7 animals-15-00065-t007:** Viscosity (cP) of the feed (n = 2) and feces (n = 12) and water content (%) of the feces (n = 6) in trial 1 (unprocessed autotrophic *C. vulgaris)* and trial 2 (PEF-processed autotrophic *C. vulgaris)*.

Trial 1										
	CON	A1	A2	A5	A10	A15	A20	Slope (linear)	CI	*p*-Value
Viscosity (cP) feed	0.99 ± 0.01	0.98 ± 0.03	0.94 ± 0.00	1.00 ± 0.18	1.09 ± 0.08	1.15 ± 0.04	1.24 ± 0.02	0.013	0.008, 0.019	<0.001
Viscosity (cP) feces	1.22 ± 0.04	1.22 ± 0.03	1.15 ± 0.07	1.23 ± 0.15	1.25 ± 0.04	1.32 ± 0.03	1.51 ± 0.02	0.014	0.009, 0.018	<0.001
Water content (%) feces	68.93 ± 2.81	69.80 ± 3.41	67.97 ± 3.18	67.91 ± 2.24	64.99 ± 2.08	60.89 ± 1.66	60.54 ± 2.88	−0.48	−0.60, −0.37	<0.001
**Trial 2**										
	**CON**	**APEF1**	**APEF2**	**APEF5**	**APEF10**	**APEF15**	**APEF20**	**Slope (linear)**	**CI**	** *p* ** **-Value**
Viscosity (cP) feed	0.97 ± 0.03	0.96 ± 0.07	1.03 ± 0.01	1.03 ± 0.03	1.15 ± 0.09	1.31 ± 0.01	1.28 ± 0.04	0.018	0.013, 0.022	< 0.001
Viscosity (cP) feces	1.08 ± 0.02	1.10 ± 0.06	1.19 ± 0.03	1.20 ± 0.02	1.26 ± 0.02	1.41 ± 0.02	1.59 ± 0.02	0.023	1.017, 1.161	< 0.001
Water content (%) feces	72.57 ± 3.06	71.03 ± 1.88	70.53 ± 2.79	67.12 ± 0.57	64.66 ± 4.19	65.27 ± 2.32	62.18 ± 2.11	−0.47	−0.59, −0.35	< 0.001

Mean values ± standard deviation and slopes of the linear correlation with *p*-value. A1–20: unprocessed autotrophic *C. vulgaris* at 1–20% inclusion level, APEF1–20: PEF-processed autotrophic *C. vulgaris* at 1–20% inclusion level, PEF: pulsed electric field, CI: confidence interval of the slope.

## Data Availability

The original contributions presented in the study are included in the article, further inquiries can be directed to the corresponding author.

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
