# Peer review of "Exploring Feed Digestibility and Broiler Performance in Response to Dietary Supplementation of Chlorella vulgaris"

_animals, 2024, doi:10.3390/ani15010065_

Round 1
Reviewer 1 Report
Comments and Suggestions for Authors
The evaluated manuscript is interesting and up-to-date. It is very well written and prepared. It extends the existing knowledge in the field of the impact of Chlorella vulgaris-containing diets on broilers. The Introductions provides sufficient background for the study, the experimental design does not raise ay doubts. The Material and Methods section contains all necessary information. The results were clearly described, presented and discussed. The manuscript can be accepted for publication after minor revision.
Comments and suggestions:
- Throughout the manuscript, citations must be corrected according to the journal’s guidelines for authors. The reference list should be adjusted accordingly.
- L. 113 and 114 – Please change to: “23 h light/1 h dark” and “18 h light/6 h dark”.
- L. 169 – There should be “cyanocobalamin”.
- L. 171 – There should be “iron (II) sulphate” and “copper (II) sulphate” (with spaces inserted).
- L. 172 - There should be “manganese (II) oxide”.
- L. 174 – There should be “selenite – selenium”.
- Table 4 – There should be “tryptophan”.
- Please change the numbering of tables. Table 4 (Composition of C. vulgaris) should be Table 2. Make all necessary changes throughout the manuscript accordingly.
- Please compare l. 148 and 151-152 with Table 2-8’ headings and unify the names of groups (A1 or A1% etc.).
- Figure 3 – Trail 2 – There is no blue line (segmented) on this figure but it is described in the text. Please, correct this.
- L. 294 – There should be “was” instead of “is”.
- L. 325-335 – The Authors should explain why ADG, FI, breast weight and color were measured only in groups given diets with APEF.
- L. 344 – Please delete “(PEF processed”.
- L. 392 – There should be “(…) it was a 12.5%...”
- L. 393 – Please delete “at 5%”.
- L. 399 – There should be “(…) and the birds would…”
- L. 416 – Please consider replacement of “Lack of…” with “Impaired nutrient digestibility…”.
- L. 429 – There should be “hydrolysis”.
- L. 462-463 – Please, delete this sentence.
- Figure 5 should be cited in the text.
- L. 473 – Please, add “(…) which was related to…”
- L. 483 – PUFA are susceptible to oxidation. Please, explain how PUFA can increase oxidative stability of breast meat.
Author Response
Q1: The evaluated manuscript is interesting and up-to-date. It is very well written and prepared. It extends the existing knowledge in the field of the impact of Chlorella vulgaris-containing diets on broilers. The Introductions provides sufficient background for the study, the experimental design does not raise any doubts. The Material and Methods section contains all necessary information. The results were clearly described, presented and discussed. The manuscript can be accepted for publication after minor revision.
A1: Thank you very much for this positive feedback.
Q2: Throughout the manuscript, citations must be corrected according to the journal’s guidelines for authors. The reference list should be adjusted accordingly.
A2: The citations and reference list is updated according to the guidelines of mdpi.
Q3: L. 113 and 114 – Please change to: “23 h light/1 h dark” and “18 h light/6 h dark”.
A3: Done as requested [line 118-119]
Q4: L. 169 – There should be “cyanocobalamin”.
A4: Done as requested [line 153]
Q5: L. 171 – There should be “iron (II) sulphate” and “copper (II) sulphate” (with spaces inserted).
A5: Done as requested [line 155]
Q6: L. 172 - There should be “manganese (II) oxide”.
A6: Done as requested [line 156]
Q7: L. 174 – There should be “selenite – selenium”.
A7: Done as requested [line 158]
Q8: Table 4 – There should be “tryptophan”.
A8: Done as requested
Q9: Please change the numbering of tables. Table 4 (Composition of C. vulgaris) should be Table 2. Make all necessary changes throughout the manuscript accordingly.
A9: Done as requested
Q10: Please compare l. 148 and 151-152 with Table 2-8’ headings and unify the names of groups (A1 or A1% etc.).
A10: Done as requested
Q11: Figure 3 – Trail 2 – There is no blue line (segmented) on this figure but it is described in the text. Please, correct this.
A11: Thank you for pointing this out. The figure was replaced by the complete and correct figure.
Q12: L. 294 – There should be “was” instead of “is”.
A12: Done as requested
Q13: L. 325-335 – The Authors should explain why ADG, FI, breast weight and color were measured only in groups given diets with APEF.
A13: Performance and meat quality evaluation would indeed also have been interesting to measure in the first trial. Unfortunately, due to practical reasons it was not possible to collect these data in the first trial.
Q14: L. 344 – Please delete “(PEF processed”.
A14: Done as requested
Q15: L. 392 – There should be “(…) it was a 12.5%...”
A15: Done as requested [line 409]
Q16: L. 393 – Please delete “at 5%”.
A16: Done as requested
Q17: L. 399 – There should be “(…) and the birds would…”
A17: Done as requested [line 416]
Q18: L. 416 – Please consider replacement of “Lack of…” with “Impaired nutrient digestibility…”.
A18: Done as requested [line 433]
Q19: L. 429 – There should be “hydrolysis”.
A19: Done as requested [line 446]
Q20: L. 462-463 – Please, delete this sentence.
A20: Done as requested
Q21: Figure 5 should be cited in the text.
A21: Done as requested [line 478]
Q22: L. 473 – Please, add “(…) which was related to…”
A22: Done as requested [line 488]
Q23: L. 483 – PUFA are susceptible to oxidation. Please, explain how PUFA can increase oxidative stability of breast meat.
A23: This was an oversight, only the following statement was kept: Furthermore, microalgae can enrich meat with omega n-3 fatty acids, since they are high in poly-unsaturated fatty acids (PUFAs) (Madeira et al., 2016). Thank you for pointing this out. [line 497-198]
Reviewer 2 Report
Comments and Suggestions for Authors
Manuscript ID: animals-3350089
Title: Correlation between Chlorella vulgaris inclusion level in broiler feed and feed digestibility
The manuscript needs some revision, because there are some aspects of the work that should be corrected and improved. Please, review the following recommendations:
- The writing lacks coherence, particularly in the introduction section, which affects the overall readability.
- In all text: Please delete all the pronouns like we, our, etc. throughout the manuscript and change the text appropriately.
- There are many Latin names of species in the full text that are not in italics. Please check lines 183, 210, 213, 223, 227 & 470.
- The introduction section is disjointed and unconnected, and each paragraph does not serve the previous or next one, so the introduction must be restructured well.
- The authors should be described the scientific rationale of the study (= what is the scientific problem?) in the Introduction section
- The hypothesis of the study should be clarified at the end of the Introduction section.
- The objective of the study should be added at the end of the introduction section.
Why Chlorella vulgaris is a sustainable protein source?
- The system of writing references within the text and in the list of references is not completely compatible with the journal system.
- Lines 11-21: All the sentences focused on the impact of demand on poultry and then finished the section with a general, out-of-place sentence about the animal.
- Lines 24-25: in this sentence "seven treatments (1%, 2%, 5%, 10%, 15% and 20% C. vulgaris)" , six or seven treatments?
- Line 37: Delete "Poultry" because you used broiler in the first word
- Lines 51-54: Are there any relevant studies that have proved this speculation? If there are, please add them.
- Lines 57-60: Are there any relevant studies that have proved this speculation? If there are, please add them.
- Lines 63-65: Revise the number in this sentence and use recent reference
- Lines 66-69: Are there any relevant studies that have proved this speculation? If there are, please add them.
- Line 93: Write the whole name before this abbreviation "PEF"
- The discussion section needs more insightful comments. The results are poorly discussed.
- Line 462: "Error! Reference source not found. "?
- Figure 5 is not mentioned in the text.
- It is preferable to shorten the "6. CONCLUSIONS"
- References used should be updated, where it is noted one reference in 2024
- Lines 620-623: Determine which is "2014a" and which is "2014b"
Comments on the Quality of English LanguageThe English could be improved to more clearly express the research
Author Response
Q1: The writing lacks coherence, particularly in the introduction section, which affects the overall readability.
A1: The introduction section was restructured with additional references to clarify some statements. This way, we hope to improve the readability. We took into account the considerations in Q4-Q8 and Q12-Q15.
Q2: In all text: Please delete all the pronouns like we, our, etc. throughout the manuscript and change the text appropriately.
A2: Done as requested.
Q3: There are many Latin names of species in the full text that are not in italics. Please check lines 183, 210, 213, 223, 227 & 470.
A3: Done as requested. All phylogenetic names are now in italic.
Q4: The introduction section is disjointed and unconnected, and each paragraph does not serve the previous or next one, so the introduction must be restructured well.
A4: The introduction section was restructured with additional references to clarify some statements. This way, we hope to improve the readability.
Q5: The authors should be described the scientific rationale of the study (= what is the scientific problem?) in the Introduction section
A5: A scientific rationale was added to the introduction section. [line 95-103]
Q6: The hypothesis of the study should be clarified at the end of the Introduction section.
A6: The hypothesis was added at the end of the Introduction section. [line 95-103]
Q7: The objective of the study should be added at the end of the introduction section. Why Chlorella vulgaris is a sustainable protein source?
A7: The objective is now pointed out in the introduction section. [line 95-103]
Q8: The system of writing references within the text and in the list of references is not completely compatible with the journal system.
A8: Citations and references are now updated according to the guidelines.
Q9: Lines 11-21: All the sentences focused on the impact of demand on poultry and then finished the section with a general, out-of-place sentence about the animal.
A9: The last lines were changed to: By exploring alternative protein sources such as microalgae, this study takes a step towards replacement of soy in poultry diets. [line 17-19]
Q10: Lines 24-25: in this sentence "seven treatments (1%, 2%, 5%, 10%, 15% and 20% C. vulgaris)" , six or seven treatments?
A10: The control treatment was missing. This line was changed to: Each trial included seven treatments (0%, 1%, 2%, 5%, 10%, 15% and 20% (%w/w on dry matter) C. vulgaris) with six replicates per treatment (three birds per replicate). [line 22-24]
Q11: Line 37: Delete "Poultry" because you used broiler in the first word
A11: Done as requested [line 37]
Q12: Lines 51-54: Are there any relevant studies that have proved this speculation? If there are, please add them.
A12: Relevant literature was added as requested. [line 51]
Q13: Lines 57-60: Are there any relevant studies that have proved this speculation? If there are, please add them.
A13: Relevant literature was added as requested. [line 55]
Q14: Lines 63-65: Revise the number in this sentence and use recent reference
A14: The introduction was restructured and we removed the statement from the section to improve readability and only retain the most important information.
Q15: Lines 66-69: Are there any relevant studies that have proved this speculation? If there are, please add them.
A15: The introduction was restructured and we removed the statement from the section to improve readability and only retain the most important information.
Q16: Line 93: Write the whole name before this abbreviation "PEF"
A16: Done as requested [line 83]
Q17: The discussion section needs more insightful comments. The results are poorly discussed.
A17: The discussion was adapted following the comments of reviewer 3.
Q18: Line 462: "Error! Reference source not found. "?
A18: Apologies, this line was removed.
Q19: Figure 5 is not mentioned in the text.
A19: Figure 5 was added to the text. [line 478]
Q20: It is preferable to shorten the "6. CONCLUSIONS"
A20: The conclusion is shortened and more to the point.
Q21: References used should be updated, where it is noted one reference in 2024
A21: Citations and references are updated according to the guidelines of mdpi.
Q22: Lines 620-623: Determine which is "2014a" and which is "2014b"
A22: This problem was solved by switching to the mdpi citation format.
Q23: The English could be improved to more clearly express the research
A23: We went through the manuscript again and English improvement was done by a native English speaker.
Reviewer 3 Report
Comments and Suggestions for Authors
The article “Correlation between Chlorella vulgaris inclusion level in broiler feed and feed digestibility” addresses very important area of poultry nutrition i.e., finding alternative dietary protein sources for poultry birds. Micro-algae (Chlorella vulgaris) cell wall disintegration can aid in digestibility leading to sustainable poultry production. However, the article needs extensive improvements. Currently, the article clearly lacks integrated approach in almost every part starting from simple summary to conclusion. Authors are strongly suggested to do following improvements
1. Go through all manuscript and improve structures of the sentenses and english grammar.
2. Follow the general rules of writing a research paper, for example, one of the major flaw is inconsistent use of abbreviations. Do not use abbreviation until explained when it appears first time in the text. Afterwards always use abbreviation. The abstract, tables, and figures are treated separate. For examples, authors write Chlorella vulgaris at few places while C vulgaris at others. In table 3 it is not italic form etc. Similalrly, use of alga, alga, microalgae, (micro)algae. Check and correct all manuscript for such inconsistencies.
3. Thoroughly read the relevant publications of well reputed journals on poultry nutrition and follow the way how different parts of the manuscript are presented. I would suggest to read following and other papers as well
A) Pokoo-Aikins A, Timmons JR, Min BR, Lee WR, Mwangi SN, Chen C. Effects of Feeding Varying Levels of DL-Methionine on Live Performance and Yield of Broiler Chickens. Animals (Basel). 2021 Sep 29;11(10):2839. doi: 10.3390/ani11102839. PMID: 34679860; PMCID: PMC8532918.
B) Badar, I. H., Jaspal, M. H., Yar, M. K., Ijaz, M., Khalique, A., Zhang, L., ... & Husnain, F. (2021). Effect of strain and slaughter age on production performance, meat quality and processing characteristics of broilers reared under tropical climatic conditions. European Poultry Science/Archiv für Geflügelkunde, (326).
Title
Since different parameters like feed consumption, meat characteristics, intestinal viscosity and others are studied, it is not simply about correlation of the inclusion levels. Therefore, title should be of broader perspective rather than to be simply correlation.
Simple summary
The authors are directed to avoid redundancy e.g., frequent use of “levels”. It also lacks the information that study was conducted with two trials.
Line 17-18: It is a vague sentense. It does not clarify at which specific level/s authors found the digestion issues and what kind of digestion issues
Line 21: It is not an environmental study, therefore, emphasizing particularly environment is not suitable.
Abstract
Presentation of results in abstract is very confusing. Results of two different trials can not be compared together. Abstract need to be extensively modified. Clearly mention which part of results belong to which trial.
Line 24-25: The unit of micro-algae inclusion level is lacking. 1%, 2%, 5%, 10%, 15% 24 and 20% C. vulgaris of what?Is it DM?
Line 26: What is meant by higher inclusion levels? Please specify
Line 28-30: Ambiguously presented results. To which trial these belong.
Introduction
Introduction lacks the necessary information and includes unnecessary one. For example, it provides extra background of PEF contrary to necessary elements like effect of microalgae in amimals. It does not provide comparative information on PEF and non-PEF micro-algae in poultry feed. Authors are suggested to improve structure and integrity of the introduction with addition of more relevant literature. Aim of the study is not clear. How Chlorella vulgaris can be beneficial to enhance nutrient digestibility?
Line 51: sustainable should be enough than more sustainable
Line 77-84: It is suggested to convince the reader by providing advantage of PEF over other mentioned techniques
Line 83: “reached” is not suitable word here. Can be replaced
Line 85-87: Need reference and types of phytochemicals present in Chlorella vulgaris
Material and methods
This part is written poorly, especially the titles of the headings and sub headings need to be revised. I would suggest to follow other publication on poultry nutrition.
The term “balance period” is not common. If authors still want to use it, then explain it.
Authors should strictly follow the flow of the material and methods. For example, broiler span should be explained once only preferably in the same sentense, then measurements, sampling, and sampling procedures by providing clear references and methodological approaches. Then analysis. Please maintain the writing flow.
Line 109: The sub heading “Two Digestibility Trials with Broilers (Trial 1 and Trial 2)” is not appropriate. Please consult relevant publications. It should be something related to study design, birds collection, birds rearing, and housing etc. Clearly differentiate between methodology of trial 1 and 2.
Line 113: Do authors mean during the first week?If yes, please correct sentense.
Line 112: Refer to the respective table of starter feed
Line 116: Replace reached with suitable word
Line 112: if authors selected FI as abbreviation, I would suggest to remove “total” from the text
Line 125: Poor sentense, please improve
Line 126: dry matter has been already abbreviated in the text, authors should use abbreviation afterwards.
Line 125-128: The references provided for analysis are not relevant. Each method, for example, Kjeldahl method (if used) for determination of crude protein, should be mentioned for each analysis. Please revise.
Line 127: What is B in crude fat-B?
Line 128: Remove apparent before fecal digestibility coefficients if you are taking FDC as abbreviation.
Line 130: No abbreviation for fecal digestibility coefficients, why?
Line 134-137: Provide references of the equations
Line 195: Not needed
Results
Results are needed to write again and their link with the relevant tables and figures need to be corrected. Agains headings are irrelevant, ambiguous and vague. Feed composition can not be the part of results in an animal nutrition study, rather it is part of material methods. There is not clear cut demarcation of results of the trial 1 and 2. Results are very poorly presented. Authors are suggested to avoid writing in text the results which are part of tables or figures.
Table 1: Vitamin and mineral premix in the footnote and within table show inconsistency
Wrong numbering of tables. Table 5 appears after table 3. Check all. Tables and Figures have many mistakes, and overstuffed with long foot notes unnecessarily. Units of the parameters are vague, e.g., %, % of what? The treatments names are ambiguosuly presented, e.g., APEF is wrongly explained in footnote of table 3. In some tables Kcal, whereas, in others it is expressed as Kcal/kg. Authors are suggested to clearly partiotion the results of trial 1 followed by trial 2 both in text and table or figures.
Discussion
Discussion need extensive improvement regarding linking findings of the current study with previous work. There is not sufficient information which can explain the rational behind changes observed in the current study. The second paragraph of discussion starting from line number 365 seems irrelevant. State clearly why NSP enzymes are discussed. Are these enzymes interfering with results of your study?
Line 354-355: Poor sentense
Line 364: Need reference
Line 365: sp italic
Line 376-378: This is overgeneralized statement, explain digestibility of what was affected?
Line 381: Digestibilities instead of digestibility417-418: It needs reference
Line 429-431: If other processes are used for cell wall degradation, then authors need to explain the advantage of PEF procedure over the other procedures
Line 440-441: PEF-treatment and NSP-enzymes why dash in between
Line 447-460: Text in these lines can be summarized into 1-2 sentenses, as reporting detailed requirement of amino acids is unnecessary.
Line 462: Problem
Line 463-468: Extensive comparison of microalgae with soybean meal reported by authors gives impression that later has been replaced in diet by the earlier which is not the case. Therefore, this comparison in discussion is not logical.
Line 474-475: Please elaborate the refered papers that how much changes in feed intake were observed in these studies. Do these support your studies? State their findings clearly
Conclusion
Conclusion needs to be re-written. It clearly lacks the take home message. Instead it is presenting over stuffed information
Comments on the Quality of English LanguageQuality of English used in article is poor and need to be extensively improved.
Author Response
Q1: The article “Correlation between Chlorella vulgaris inclusion level in broiler feed and feed digestibility” addresses very important area of poultry nutrition i.e., finding alternative dietary protein sources for poultry birds. Micro-algae (Chlorella vulgaris) cell wall disintegration can aid in digestibility leading to sustainable poultry production. However, the article needs extensive improvements. Currently, the article clearly lacks integrated approach in almost every part starting from simple summary to conclusion. Authors are strongly suggested to do following improvements
A1: Thank you for the overall feedback.
Q2: Go through all manuscript and improve structures of the sentenses and english grammar.
A2: We went through the manuscript again and English improvement was done by a native English speaker.
Q3: Follow the general rules of writing a research paper, for example, one of the major flaw is inconsistent use of abbreviations. Do not use abbreviation until explained when it appears first time in the text. Afterwards always use abbreviation. The abstract, tables, and figures are treated separate. For examples, authors write Chlorella vulgaris at few places while C vulgaris at others. In table 3 it is not italic form etc. Similalrly, use of alga, alga, microalgae, (micro)algae. Check and correct all manuscript for such inconsistencies.
A3: Thank you for pointing this out. The text was improved and consistency was taken into account. Algae was changed to microalgae. Abbreviations were used and italic notation of the species names were used.
Q4: Thoroughly read the relevant publications of well reputed journals on poultry nutrition and follow the way how different parts of the manuscript are presented. I would suggest to read following and other papers as well
- A) Pokoo-Aikins A, Timmons JR, Min BR, Lee WR, Mwangi SN, Chen C. Effects of Feeding Varying Levels of DL-Methionine on Live Performance and Yield of Broiler Chickens. Animals (Basel). 2021 Sep 29;11(10):2839. doi: 10.3390/ani11102839. PMID: 34679860; PMCID: PMC8532918.
- B) Badar, I. H., Jaspal, M. H., Yar, M. K., Ijaz, M., Khalique, A., Zhang, L., ... & Husnain, F. (2021). Effect of strain and slaughter age on production performance, meat quality and processing characteristics of broilers reared under tropical climatic conditions. European Poultry Science/Archiv für Geflügelkunde, (326).
A4: Thank you very much for the suggestions. We attempted to improve the structure of this study. However, the articles mentioned above are based on performance trials which required a different design than our digestibility trials. We also report a part on performance, however this was not the main focus of this study and therefore this is mentioned together with additional data on meat quality at the end for completion. The insights in growth and meat color were significant and we were convinced that this information would complete the overall study. Nevertheless, the study design was suited to make conclusions on digestibility and therefore we opt to mention these insights first.
Title
Q5: Since different parameters like feed consumption, meat characteristics, intestinal viscosity and others are studied, it is not simply about correlation of the inclusion levels. Therefore, title should be of broader perspective rather than to be simply correlation.
A5: The title was adjusted and more related to the content of the paper: Exploring the Impact of Chlorella vulgaris Inclusion Levels in Broiler Feed on Digestibility and Performance
Simple summary
Q6: The authors are directed to avoid redundancy e.g., frequent use of “levels”. It also lacks the information that study was conducted with two trials.
A6: Changes were made as requested:
This study explored the potential of microalga Chlorella (C.) vulgaris as a sustainable protein source in broiler feed. Two trials were conducted to evaluate broilers' digestion of feeds containing varying levels of C. vulgaris. This alga was tested in two forms: unprocessed biomass and biomass processed using a pulsed electric field (PEF), a technique designed to disrupt cell walls. Broiler diets were supplemented C. vulgaris in amounts ranging from 1% - 20%. The results showed decreasing digestibility of protein, fat, and energy as microalgae inclusion levels increased. However, treating C. vulgaris with PEF led to improved digestibility of the feed. By exploring alternative protein sources such as microalgae, this study takes a step towards replacement of soy in poultry diets.
Q7: Line 17-18: It is a vague sentense. It does not clarify at which specific level/s authors found the digestion issues and what kind of digestion issues
A7: This sentence was indeed vague and is now removed from the simple summary. In the abstract, the specific levels are now mentioned.
Q8: Line 21: It is not an environmental study, therefore, emphasizing particularly environment is not suitable.
A8: This statement is also removed from the simple summary.
Abstract
Q9: Presentation of results in abstract is very confusing. Results of two different trials can not be compared together. Abstract need to be extensively modified. Clearly mention which part of results belong to which trial.
A9: Done as requested:
This study evaluated the digestibility of autotrophic Chlorella (C.) vulgaris in 252 male broilers (Ross 308), comparing unprocessed biomass (trial 1) and pulsed electric field (PEF) processed biomass (trial 2) at inclusion levels up to 20%. Each trial included seven treatments (0%, 1%, 2%, 5%, 10%, 15% and 20% (%w/w on dry matter) C. vulgaris) with six replicates (three birds per replicate) per treatment. Data were analyzed using linear, quadratic, and broken-line models. Control feeds without microalgae inclusion achieved a crude protein digestibility of 82.04 ± 1.42% (trial 1) and 81.63 ± 1.90% (trial 2), while feed with 20% non-processed microalgae inclusion only had a protein digestibility of 66.96 ± 1.16% (trial 1) and feed with PEF processed microalgae at 20% had a protein digestibility of 72.75 ± 0.34% (trial 2). In general, increasing inclusion levels of C. vulgaris impaired nutrient digestibility, significantly reducing crude protein, crude fat, gross energy, and crude ash digestibility (P < 0.001). Broken-line models identified critical inclusion thresholds beyond which digestibility declined significantly, i.e., at 10% for crude protein, 12.53% for crude fat, and 9.26% for gross energy in unprocessed microalgae feeds (trial 1). For PEF processed microalgae, only a broken line fit was obtained for gross energy, with a breakpoint at 5% (trial 2). This research advances the exploration of sustainable protein alternatives, highlighting the potential of microalgae in broiler feed and the benefits of processing methods such as PEF to enhance nutrient utilization.
Q10: Line 24-25: The unit of micro-algae inclusion level is lacking. 1%, 2%, 5%, 10%, 15% 24 and 20% C. vulgaris of what?Is it DM?
A10: The unit was added: 0%, 1%, 2%, 5%, 10%, 15% and 20% (%w/w on dry matter) C. vulgaris). [line 23]
Q11: Line 26: What is meant by higher inclusion levels? Please specify
A11: The statement was changed as follows: ). In general, increasing inclusion levels of C. vulgaris impaired nutrient digestibility, significantly reducing crude protein, crude fat, gross energy, and crude ash digestibility (P < 0.001). [line 28-30]
Q12: Line 28-30: Ambiguously presented results. To which trial these belong.
A12: After every statement, the trial to which it belongs is now mentioned.
Introduction
Q13: Introduction lacks the necessary information and includes unnecessary one. For example, it provides extra background of PEF contrary to necessary elements like effect of microalgae in animals. It does not provide comparative information on PEF and non-PEF micro-algae in poultry feed. Authors are suggested to improve structure and integrity of the introduction with addition of more relevant literature. Aim of the study is not clear. How Chlorella vulgaris can be beneficial to enhance nutrient digestibility?
A13: Thank you very much for this remark. We improved the structure of the introduction by pointing out the most important statements and remove background information. PEF can improve the digestibility of Chlorella by disrupting its rigid cell wall. That way, nutrients will become easier available for the broilers. However, the last question of your remark is not our research goal: we wanted to test Chlorella as a feed ingredient and we wanted to see how much this limits the digestibility of the feeds. We did not expect Chlorella to be beneficial for nutrient digestibility, mainly due to its complex structured cell wall. We also added more literature in the introduction of as was already pointed out by the previous reviewers.
Q14: Line 51: sustainable should be enough than more sustainable
A14: Done as requested. [line 49]
Q15: Line 77-84: It is suggested to convince the reader by providing advantage of PEF over other mentioned techniques
A15: This is now explained more deeply: PEF is less energy consuming than other techniques. Furthermore, PEF only disrupts cell walls instead of completely breaking them, which is favorable to avoid oxidation of nutrients inside the cells. [line 90-94]
Q16: Line 83: “reached” is not suitable word here. Can be replaced
A15: This part was removed from the introduction.
Q16: Line 85-87: Need reference and types of phytochemicals present in Chlorella vulgaris
A16: The following was added: due to the presence of bio-active compounds with potential health-promoting effects such as poly-unsaturated fatty acids (PUFAs) and pigments such as β-carotene, lutein and astaxanthin that can serve as antioxidants. Furthermore, C. vulgaris contains vitamins (e.g., B1, B2, B3, B5, B6, B7, B9, B12, C, E and A) and minerals (e.g., Na, K, Ca, Mg, P, Cr, Cu, Zn, Mn, Se, I, Fe) that have potential health benefits for poultry (Safi, 2014). [line 62-69]
Material and methods
Q17: This part is written poorly, especially the titles of the headings and sub headings need to be revised. I would suggest to follow other publication on poultry nutrition.
A17: The m&m sections is now restructured and subheadings and headings are changed to make the flow more clear.
Q18: The term “balance period” is not common. If authors still want to use it, then explain it.
A18: The term is now explained in the text. [line 125-126]
Q19: Authors should strictly follow the flow of the material and methods. For example, broiler span should be explained once only preferably in the same sentense, then measurements, sampling, and sampling procedures by providing clear references and methodological approaches. Then analysis. Please maintain the writing flow.
A19: We restructured the flow of the materials and methods sections as requested.
Q20: Line 109: The sub heading “Two Digestibility Trials with Broilers (Trial 1 and Trial 2)” is not appropriate. Please consult relevant publications. It should be something related to study design, birds collection, birds rearing, and housing etc. Clearly differentiate between methodology of trial 1 and 2.
A20: The heading was changed as requested. [line 114]
Q21: Line 113: Do authors mean during the first week?If yes, please correct sentense.
A22: Indeed, change is done as requested. [line 119]
Q23: Line 112: Refer to the respective table of starter feed
A23: Done as requested. [line 118]
Q24: Line 116: Replace reached with suitable word
A24: Done as requested. (Reached was removed).
Q25: Line 112: if authors selected FI as abbreviation, I would suggest to remove “total” from the text
A25: Done as requested.
Q26: Line 125: Poor sentense, please improve
A26: Done as requested. [line 184-185]
Q27: Line 126: dry matter has been already abbreviated in the text, authors should use abbreviation afterwards.
A27: Done as requested.
Q28: Line 125-128: The references provided for analysis are not relevant. Each method, for example, Kjeldahl method (if used) for determination of crude protein, should be mentioned for each analysis. Please revise. Line 127: What is B in crude fat-B?
A28: The references that are mentioned are the methods that are used in our lab which is certified for analysis of feed samples following the ISO standard protocols. Crude fat B is a protocol that first hydrolyses the fatty acids with HCl and then extract them with petroleum ether. Crude fat A is a protocol where only extraction with petroleum ether is done. [line 184-185]
Q29: Line 128: Remove apparent before fecal digestibility coefficients if you are taking FDC as abbreviation.
A29: Apparent was added to the abbreviation: aFDC
Q30: Line 130: No abbreviation for fecal digestibility coefficients, why?
A30: We used aFDC for apparent fecal digestibility coefficients, because we used it in the Equations. For specific digestibility coefficients no further abbreviations were used to keep the further results section easy to understand.
Q31: Line 134-137: Provide references of the equations. Line 195: Not needed
A31: A reference article for the equations was added. Line 195: Statistical significance levels is important for interpretation of the results. Therefore, we would like to keep this sentence in the text. [line 191]
Results
Q32: Results are needed to write again and their link with the relevant tables and figures need to be corrected. Agains headings are irrelevant, ambiguous and vague. Feed composition can not be the part of results in an animal nutrition study, rather it is part of material methods. There is not clear cut demarcation of results of the trial 1 and 2. Results are very poorly presented. Authors are suggested to avoid writing in text the results which are part of tables or figures.
A32: The article’s main objective was to investigate the digestibility of the feeds. Therefore, the headings indicate all digestibility parameters (protein, fat, ash, energy). These are the main findings of the study and are therefore explained under separate headings. The last headings concerning growth, meat quality and viscosity are some extra parameters that were investigated to make the discussion more profound. Therefore, these headings are put at the end of the paper. The subheading ‘analyzed feed composition’ was replaced to the m&m section. The difference between trial 1 and trial 2 is always clearly indicated in the text, as well as in the tables, the table captions and the figures and figure captions. We added the relevant trial in all headings to make it more clear.
Q33: Table 1: Vitamin and mineral premix in the footnote and within table show inconsistency
A33: Changed as requested.
Q34: Wrong numbering of tables. Table 5 appears after table 3. Check all. Tables and Figures have many mistakes, and overstuffed with long foot notes unnecessarily. Units of the parameters are vague, e.g., %, % of what? The treatments names are ambiguosuly presented, e.g., APEF is wrongly explained in footnote of table 3. In some tables Kcal, whereas, in others it is expressed as Kcal/kg. Authors are suggested to clearly partiotion the results of trial 1 followed by trial 2 both in text and table or figures.
A34: The numbering of the tables was changed. Trial 1 and trial 2 is always clearly indicated in the tables and in the captions. In the figures, trial was is one the left and trial 2 on the right (which is also clearly indicated on top of the figures and in the captions.) The footnotes of the tables are now adjusted to make the abbreviations more clear. Kcal was changed to Kcal/kg. % is (%w/w on dry matter) in the feed. This was added in the caption. The footnotes are indeed long; however, for interpretation it is important to report all the model parameters and confidence intervals.
Discussion
Q35: Discussion need extensive improvement regarding linking findings of the current study with previous work. There is not sufficient information which can explain the rational behind changes observed in the current study. The second paragraph of discussion starting from line number 365 seems irrelevant. State clearly why NSP enzymes are discussed. Are these enzymes interfering with results of your study?
A35: The second paragraph discusses the influence the rigid cell wall might have on the digestibility and compared this to Spirulina, which does not have a complex cell wall. Therefore, we think the paragraph is still interesting. In the current study, NSP enzymes are added, as is done in standard chicken diets. This might also have an impact on the digestibility because these enzymes might be able to break the polysaccharide cell wall of C. vulgaris. NSP enzymes include xylanase, which will have an effect on the C. vulgaris cell wall.
Q36: Line 354-355: Poor sentence
A36: The sentence was rewritten. [line 368-371
Q37: Line 364: Need reference
A37: References is added. [line 371]
Q38: Line 365: sp italic
A38: According to notation recommendations for scientific names; sp. or spp. should not be written in italic.
Q39: Line 376-378: This is overgeneralized statement, explain digestibility of what was affected?
A39: The following was added: The addition of enzymes that can break down the rigid cell wall of C. vulgaris may offer a solution to improve the availability and digestibility of nutrients within the cells. [line 387-389]
Q40: Line 381: Digestibilities instead of digestibility Line 417-418: It needs reference
A40: Done as requested. [line 398], [line 436]
Q41: Line 429-431: If other processes are used for cell wall degradation, then authors need to explain the advantage of PEF procedure over the other procedures
A41:An explanation was added in the text. [line 453-457]
Q42: Line 440-441: PEF-treatment and NSP-enzymes why dash in between
A42: The dashes were removed.
Q43: Line 447-460: Text in these lines can be summarized into 1-2 sentenses, as reporting detailed requirement of amino acids is unnecessary.
A43: The text is summarized and numbers are removed from the discussion section. [472-474]
Q44: Line 462: Problem
A44: The was a problem with the reference, this is now solved. [line 479]
Q45: Line 463-468: Extensive comparison of microalgae with soybean meal reported by authors gives impression that later has been replaced in diet by the earlier which is not the case. Therefore, this comparison in discussion is not logical.
A45: The soybean was indeed not replaced 1 on 1 by the microalgae. However, for example, in the 20% feed, there is no longer soybean formulated in the feed because of the high protein amounts of Chlorella. Therefore, a comparison with their amino acid profile is indeed required.
Q46: Line 474-475: Please elaborate the refered papers that how much changes in feed intake were observed in these studies. Do these support your studies? State their findings clearly
A46: The references were only added because they mention palatability as an explanation for reduced feed intake. No feed intake in broilers was reported in these papers. (One paper is about lambs, the other is a review paper). However, they both state that microalgae inclusion could reduce palatability. Furthermore, articles on high inclusion levels of microalgae are scarce, therefore there is no feed intake data to compare with.
Conclusion
Q47: Conclusion needs to be re-written. It clearly lacks the take home message. Instead it is presenting over stuffed information
A47: The conclusion is now shortened and more to the point:
This study examined the digestibility of autotrophic C. vulgaris in broilers by comparing unprocessed and PEF processed microalgae at inclusion levels up to 20%. Digestibility was reduced at increasing inclusion levels. PEF processing of autotrophic C. vulgaris mitigated these effects on digestibility.
Broken-line models identified critical thresholds for unprocessed C. vulgaris: 10% for crude protein, 12.53% for crude fat, and 9.26% for gross energy. No significant breakpoints were observed for PEF processed microalgae except for gross energy, indicating a more gradual decline in digestibility.
Limiting C. vulgaris inclusion to no more than 10% of the diet is advisable, although digestibility still falls short compared to soybean meal, particularly for crude protein and ash. While C. vulgaris is a promising sustainable protein source, further research into processing methods and feed formulations (e.g., adding enzymes) is needed to enhance its suitability for broiler diets.
Q48: Quality of English used in article is poor and need to be extensively improved.
A48: We went through the manuscript again and English improvement was done by a native English speaker.
Round 2
Reviewer 2 Report
Comments and Suggestions for Authors
The authors made all required modifications.
Comments on the Quality of English LanguageThe English could be improved to more clearly express the research.
Author Response
Reviewer 2:
Q1: The authors made all required modifications.
A1: Thank you for the feedback.
Q2: The English could be improved to more clearly express the research.
A2: The text was read and improved by a native English speaker, as mentioned in the acknowledgements [Line 529-530].
Academic editor:
Q1: The authors need to clarify the experimentation design. Were pens employed? If so what were their sizes and what were the number of birds per pen. If not a pen study, details of what was done are needed.
A1: The M&M section mentioned that digestibility units were used, housing 3 birds per unit. For each treatment, 6 units (replicates) were used. In total a number of 42 digestibility units per trial were used (7 treatments (including a control treatment) in 6 replicates).
Following information is added now: the digestibility units have a size of L: 0.50 m, W: 0.40 m, H: 0.35 m. [Line 123].
The digestibility units are enriched with shiny objects. The digestibility units are equipped with a nipple drinker and a feeder. The chickens sit on grids and their feces are collected in collection boxes underneath these grids.
The first seventeen days (before the birds go into the digestibility units), all the birds are group-housed on a solid floor covered with wood shavings provided with drink nipples and a basal starter feed ad libitum.
Q2: I'm unclear also why the two overall treatments were examined in two separate trials (comparing untreated algae and algae processed using a pulsed electric field). I would much prefer the authors conduct a trial (say using high concentrations) comparing the untreated algae and algae processed using a pulsed electric field.
A2: This is indeed a good suggestion. However, this is mainly done due to practical reasons, as only 42 digestibility units are available at our institution. Therefore, we performed two trials, one with untreated biomass and one with PEF biomass. Since we applied 7 treatments (in 6 replicates), we needed 42 digestibility units. The choice of 7 inclusion levels per type of biomass was needed to be able to make an accurate regression on the data. Especially because we wanted to range between 0% and 20% microalgae. This was needed because we wanted to look into the possibility of using C. vulgaris as a protein source. Therefore, going up to 20% was needed and 7 treatments were needed to span this range. We first conducted the trial with untreated biomass. As we found in previous trials that PEF treatment could be beneficial, we certainly wanted to repeat the trial with PEF processed biomass.
However, the results of both trials were not compared statistically, as this is not allowed. Only the numerical values were compared. To do this, we checked if the control treatments showed similar results (as exactly the same diet under the same circumstances was given in both trial 1 and trial 2 as a control group). In trial 1, the control feed had for example a crude protein digestibility of 82.04 ± 1.42% and in trial 2 this was 81.63 ± 1.90%, for crude fat, we obtained 88.01 ± 2.39% in trial 1 and trial 2 had 89.71 ± 1.39%. This gives an indication that the conditions in both trials were similar, and therefore both trials were (although numerically) compared.
Reviewer 3 Report
Comments and Suggestions for Authors
I acknowledge the efforts of the authors to improve the quality of the manuscript entitled “Exploring the Impact of Chlorella vulgaris Inclusion Levels in Broiler Feed on Digestibility and Performance”. Successful efforts have been to improve english and structure of the sentences. However, still there are many discrepencies which need to be addressed.
Here is the detail.
Throughout the manuscript, the reader presumes that digestibility of the microalgae is going to be discussed, but in results or discussion part, digestibility of nutrients is the focus. It is evident from the multiple time use of “digestibility of microalgae”. This confusion must be settled.
Authors multiple time used the word “soy”. This is the main ingredient which is going to be replaced by the microalgae processed or non processed, but it literally does not clears its meaning whether the authors mean soybean meal or soy hulls or soy oil. In respective table the replacement is not clear.
The references of the text are not adequate at least where necessary, for example, line 69-70 and 118-119 and 368 need reference to which authors are following
Some sentenses do not clear their meaning, for example, line 15-16: To which level of microalgae, deigestibility was decreased?
The result part summarizes only the findings related to digestibility, whereas, other results, for example, of meat characters and viscosity etc.
Many sentenses are still vague and grammatically poor, for example, line number 116-117
Data in the tables is still poorly presented. A total of 4 tables are specified just for “composition”. Secondly, treatment names are confusing. Authors are suggested to make it simple, for example keep same names of the treatment by clearly differentiating between both trials. In headings of results why authors use A and APEF. The title of the table 1 is much confusing.
Despite of much emphasis in previous revision on use of inconsistent abbreviations, there are still many inconsistencies, for example, FI (feed intake) at line 125, 127, 181, and NSP in discussion part.
What is (g/d/a) in tables
Conclusion states that authors compared the effect of unprocessed and PEF processed microalgae, which were actually studied in two different experiments. Therefore, this claim is wrong
Author Response
Reviewer 3:
Q1: I acknowledge the efforts of the authors to improve the quality of the manuscript entitled “Exploring the Impact of Chlorella vulgaris Inclusion Levels in Broiler Feed on Digestibility and Performance”. Successful efforts have been to improve english and structure of the sentences. However, still there are many discrepencies which need to be addressed.
A1: Thank you for this overall feedback.
Q2: Throughout the manuscript, the reader presumes that digestibility of the microalgae is going to be discussed, but in results or discussion part, digestibility of nutrients is the focus. It is evident from the multiple time use of “digestibility of microalgae”. This confusion must be settled.
A2: This is indeed a justified remark. We tried to state it more clear in the introduction section that we investigate the effects of C. vulgaris on overall macronutrient digestibility and not the digestibility of C. vulgaris as such. We added this in lines 60-62, 73-74, 99-100, 509-511.
The title is changed to: Exploring the Impact of Chlorella vulgaris Inclusion Levels in Broiler Diets on Feed Digestibility and Performance
Q3: Authors multiple time used the word “soy”. This is the main ingredient which is going to be replaced by the microalgae processed or non processed, but it literally does not clears its meaning whether the authors mean soybean meal or soy hulls or soy oil. In respective table the replacement is not clear.
A3: We replaced the word ‘soy’ by ‘soybean’ in the text to clarify this. In Table 1 with the feed composition, it is clear that soybean, soybean meal and soy oil is not included anymore in the feed with 20% microalgae, due to the high levels of protein (and fat) in microalgae. We replaced the ‘-‘ in the table by ‘0’ to make it more clear that this ingredients is not included in the feed. [Lines 18, 42, 46, 101-102 and Table 1]. Indeed as you mentioned, in the 20% feed, soybean, soybean meal and soy oil is completely replaced by microalgae. Therefore, we chose now to use the word ‘soybean’ in the text, as these are all soybean fractions.
|
|
Starter |
F0 |
F1 |
F2 |
F20 |
|
Ingredient (%) |
|
|
|
|
|
|
Wheat |
56.17 |
51.67 |
52.45 |
53.23 |
62.37 |
|
Wheat bran |
0 |
0 |
0 |
0 |
10.00 |
|
Maize |
10.00 |
10.00 |
10.00 |
10.00 |
1.32 |
|
Soybean |
5.00 |
7.26 |
7.00 |
7.00 |
0 |
|
Soybean meal (48 % CP) |
22.99 |
22.48 |
21.32 |
19.94 |
0 |
|
Soy oil |
0.89 |
1.00 |
1.00 |
1.00 |
0 |
Q4: The references of the text are not adequate at least where necessary, for example, line 69-70 and 118-119 and 368 need reference to which authors are following
A4: Line 69-70 states: The most commonly-studied inclusion levels in poultry feed were between 0.01% and 2%, occasionally up to 7.5%. Higher inclusion levels were rarely evaluated [7,11,12].
References 7, 11 and 12 indicate three studies that investigate low amounts of microalgae to prove my point. I added another reference to a review paper I recently published that covers all studies on health promoting effects of microalgae on chickens. There, it is also clear that mostly low amounts are investigated [Line 70]. Line 118-119 does not include references as this is part of the M&M section: The first seventeen days, they were group-housed on a solid floor covered with wood shavings and were fed a basal starter diet (Table 1). Line 368 is part of the heading of Table 8.
Q5: Some sentences do not clear their meaning, for example, line 15-16: To which level of microalgae, digestibility was decreased?
A5: We showed that there is a negative linear (and quadratic, sometimes segmented) relationship between microalgae in the feeds and feed digestibility. This means that increasing microalgae in feeds, decreases its digestibility. There is no specific inclusion levels where this effect starts. However, the segmented (or broken line) models indicated that there are some critical inclusion levels after which digestibility even decreases in a steeper way than before these critical inclusion levels.
Furthermore, the abstracts gives more information about the critical inclusion levels we found by fitting a broken-line model. We like to keep the ‘simple summary’ shorter and to the point. More information (e.g. specific inclusion levels) is already stated in the abstract and of course further on in the results and discussion sections.
Q6: The result part summarizes only the findings related to digestibility, whereas, other results, for example, of meat characters and viscosity etc.
A6: The results sections also included other parameters, under 3.5 and 3.6:
3.5. Growth, Feed Intake, Breast Weight and Meat Color in Trial 2
3.6. Viscosity and Water Content of the Feces in Trials 1 and 2
Q7: Many sentences are still vague and grammatically poor, for example, line number 116-117
A7:The sentence was changed to: A total of 252 Ross 308 one-day-old male broilers were purchased from a commercial hatchery (126 per trial) (Belgabroed, Merksplas, Belgium). [Line 115-116].
Q8: Data in the tables is still poorly presented. A total of 4 tables are specified just for “composition”. Secondly, treatment names are confusing. Authors are suggested to make it simple, for example keep same names of the treatment by clearly differentiating between both trials. In headings of results why authors use A and APEF. The title of the table 1 is much confusing.
A8: We indeed show Table 1 with the feed ingredients and calculated nutrient composition, Table 2 with the C. vulgaris compositions and Table 3 with the analyzed nutrient compositions. (The previous Table 3 and 4 are now combined in Table 3 to reduce the number of tables). The other table names and references were also adjusted. However, since this a digestibility study, we believe that feed compositions is of high importance for the interpretation of the results. We use both A and APEF to make it clear that in trial 1 we used A (untreated Autotrophic C. vulgaris) and in trial 2 we used APEF (PEF treated Autotrophic C. vulgaris). These abbreviations are clearly explained in lines 129-134. The title of table 1 includes the mixtures of the formulated feeds (F0, F1, F2 and F20) to obtain our test feeds (A and APEF1-20).
Q9: Despite of much emphasis in previous revision on use of inconsistent abbreviations, there are still many inconsistencies, for example, FI (feed intake) at line 125, 127, 181, and NSP in discussion part.
A9: The abbreviations were now adjusted. [Lines 126, 127, 180, 390-391].
We checked all abbreviations (C. vulgaris, PEF, DM, PUFA, CON, A, APEF, FI, ADFI, ADG, CP, aFDC, AA and NSP) and mentioned them once fully in simple summary, abstract and in the main text. Further on, we used the abbreviations. In captions of tables and figures all abbreviations are always explained.
Q10: What is (g/d/a) in tables
A10: This means grams (g) per day (d) per animal (a). This is indicated in lines 351-352.
Q11: Conclusion states that authors compared the effect of unprocessed and PEF processed microalgae, which were actually studied in two different experiments. Therefore, this claim is wrong
A11: This statement is now removed from the conclusion and changed to: This study examined the digestibility of autotrophic C. vulgaris in broilers by adding unprocessed and PEF processed microalgae in the feed at inclusion levels up to 20% [Lines 509-511].
Round 3
Reviewer 3 Report
Comments and Suggestions for Authors
Thanks for the authors to incorporate the changes. The quality of the manuscript has been sufficiently improved. However, authors are suggested to consider the following
1. I suggest title to be “Exploring the digestibility and performance of broilers in response to dietary supplementation of Chlorella vulgaris”
2. Is it Soybean meal or Soybean which need to be included in text? I think it is Soybean meal if authors are going to stress protein source.
3. In abstract, please mention all results in addition to those of digestibility results
4. At line 511, complete the sentense by adding “inclusion levels of C. vulgaris”.
Author Response
Q1: Thanks for the authors to incorporate the changes. The quality of the manuscript has been sufficiently improved. However, authors are suggested to consider the following
A1: Thank you very much for this feedback.
Q2: I suggest title to be “Exploring the digestibility and performance of broilers in response to dietary supplementation of Chlorella vulgaris”
A2: Thank you for your suggestion. We changed the title to: “Exploring feed digestibility and broiler performance in response to dietary supplementation of Chlorella vulgaris”
Q3: Is it Soybean meal or Soybean which need to be included in text? I think it is Soybean meal if authors are going to stress protein source.
A3: Thank you for this remark. We changed ‘soybean’ to ‘soybean meal’ in the test, as we are indeed discussing the replacement of a protein source and soybean meal contains 48% crude protein. However, it is important to keep in mind that in the 20% Chlorella diets, also soybean and soy oil is no longer included.
Q4: In abstract, please mention all results in addition to those of digestibility results
A4: This is indeed an important remark. The following was added to the abstract: Furthermore, a significant linear decrease in body weight (BW) (P < 0.001), average daily gain (ADG) (P < 0.001), average daily feed intake (ADFI) (P = 0.006) and relative and absolute breast filet weight was observed as microalgae inclusion level increased (trial 2). Color parameters also changed significantly with increasing microalgae inclusion level: L* showed a significant linear decrease (P = 0.029), b* and a* showed a significant linear increase (P < 0.001) (trial 2).
Q5: At line 511, complete the sentense by adding “inclusion levels of C. vulgaris”.
A5: Done as requested.